# Wavelet regression and additive models for irregularly spaced data

**Asad Haris**[*]
Department of Biostatistics
University of Washington
Seattle, WA 98195
`aharis@uw.edu`

**Noah Simon**
Department of Biostatistics
University of Washington
Seattle, WA 98195
`nrsimon@uw.edu`

**Ali Shojaie**
Department of Biostatistics
University of Washington
Seattle, WA 98195
`ashojaie@uw.edu`

## Abstract

We present a novel approach for nonparametric regression using wavelet basis functions. Our proposal, `waveMesh`, can be applied to non-equispaced data with sample size not necessarily a power of 2. We develop an efficient proximal gradient descent algorithm for computing the estimator and establish adaptive minimax convergence rates. The main appeal of our approach is that it naturally extends to additive and sparse additive models for a potentially large number of covariates. We prove minimax optimal convergence rates under a weak compatibility condition for sparse additive models. The compatibility condition holds when we have a small number of covariates. Additionally, we establish convergence rates for when the condition is not met. We complement our theoretical results with empirical studies comparing `waveMesh` to existing methods.

## 1 Introduction

We consider the canonical task of estimating a regression function, $f$, from observations $\{(\boldsymbol{x}_i, y_i) : i = 1, \ldots, n\}$, with $\boldsymbol{x}_i \in [0, 1]^p$, $y_i \in \mathbb{R}$ and $y_i = f(\boldsymbol{x}_i) + \varepsilon_i$ $(i = 1, \ldots, n)$, where $\varepsilon_i$ are independent, mean 0, sub-Gaussian random variables. A popular approach for estimating $f$ is to use linear combinations of a pre-specified set of *basis functions*, e.g., polynomials, splines [Wahba, 1990], wavelets [Daubechies, 1992], or other systems [Čencov, 1962]. The weights, or coefficients, in such a linear combination are often determined using some form of penalized regression. In this paper, we focus on estimators that use *wavelets*. Wavelet-based estimators have compelling theoretical properties. However, a number of issues have limited their adaptation in many non-parametric applications. The approach proposed in this paper overcomes these issues. Throughout the paper, we assume basic knowledge of wavelet methods though some key points will be reviewed. For a detailed introduction to wavelets, see books by Daubechies [1992], Percival and Walden [2006], Vidakovic [2009], Nason [2010], Ogden [2012].

Wavelets are a system of orthonormal basis functions for $L^2([0, 1])$. Wavelets are popular for representing functions because they allow *time and frequency localization* [Daubechies, 1990] as opposed to, say, Fourier bases, which allow only frequency localization. Additionally, wavelet-based

---

[*]Mailing address: Box 357232, University of Washington, Seattle, WA 98195-7232

methods are computationally efficient. The main ingredient of wavelet regression is the discrete wavelet transform (DWT) and its inverse (IDWT) which can be computed in $O(n)$ operations [Mallat, 1989]. Unfortunately, traditional wavelet methods require stringent conditions on the data, specifically that $x_i = i/n$ with $n = 2^J$ for some integer $J$. This is not a problem in many signal processing applications with regularly sampled signals; however, in general non-parametric regression, this condition will rarely be satisfied. A simple solution for general data types is to ignore irregular spacing of data [Cai and Brown, 1999, Sardy et al., 1999] and/or artificially extend the signal such that $n = 2^J$ [Strang and Nguyen, 1996, Ch. 8]. Other solutions include transformations [Cai and Brown, 1998, Pensky and Vidakovic, 2001] or interpolation [Hall and Turlach, 1997, Kovac and Silverman, 2000, Antoniadis and Fan, 2001] of the data to a regular grid of size $2^J$. The literature on univariate wavelet methods is quite extensive and cannot be adequately discussed within this manuscript. In contrast, the literature on wavelet methods for multiple covariates is rather limited, particularly when the number of covariates is large.

For the multivariate settings with $\boldsymbol{x}_i \in [0, 1]^p$ for $p \geq 2$, we consider estimating an additive model, i.e., $\widehat{f}(\boldsymbol{x}_i) = \sum_j \widehat{f}_j(x_{ij})$. Additive models naturally extend linear models to capture non-linear conditional relationships, while retaining some interpretability; they also do not suffer from the *curse of dimensionality*. Despite these benefits, wavelet-based additive models have received limited attention. This is most likely because data with multiple covariates are rarely available on a regular grid of size $n = 2^J$. Sardy and Tseng [2004] fit additive wavelet models by treating the data as if regularly spaced; however, they do not discuss the case when $n$ is not a power of 2. A number of proposals transform the data to a regular grid [Amato and Antoniadis, 2001, Zhang and Wong, 2003, Grez and Vidakovic, 2018]. However, to do this, the density of the covariates must be estimated, which unnecessarily invokes the curse of dimensionality. In addition, to the best of our knowledge, there are no wavelet-based methods for fitting additive models in high dimensions (when $p > n$) that induce sparsity, i.e., for many $j$, give a solution with $\widehat{f}_j \equiv 0$.

In this paper, we give a simple proposal that effectively extends wavelet-based methods to non-parametric modeling with a potentially large number of covariates. We present an interpolation-based approach for dealing with irregularly spaced data when $n$ is not necessarily a power of 2. However, unlike existing interpolation methods, we do not transform the raw data $(\boldsymbol{x}_i, y_i)$. As a result, our method naturally extends to additive and sparse additive models. We also propose a penalized estimation framework to induce sparsity in high dimensions. We develop a proximal gradient descent method for computation of our estimator, which leverages fast algorithms for DWT and sparse matrix multiplication. Furthermore, we establish adaptive minimax convergence rates (up to a $\log n$ factor) similar to that of existing wavelet methods for regularly spaced data. We also establish convergence rates for our (sparse) additive proposal for a potentially large number of covariates. We discuss an extension of our proposal to general convex loss functions, and a weighted variation of our penalty which exhibits improved performance.

In Section 2 we present our univariate, additive and sparse additive proposals. The univariate case ($p = 1$) is mainly presented to motivate our proposal. We also present our main algorithm for computing the estimator. We establish convergence rates of our estimators in Section 3, and present empirical studies in Section 4. Concluding remarks are given in Section 5.

## 2 Methodology

### 2.1 Short background on wavelets

We begin with a quick review of wavelet methods for nonparametric regression covering 3 main ingredients: (1) wavelet basis functions, (2) the discrete wavelet transform (DWT) and, (3) shrinkage.

First, *wavelets* are a system of orthonormal basis functions for $L^2([0, 1])$ or $L^2(\mathbb{R})$. The bases are generated by translations and dilations of special functions $\phi(\cdot)$ and $\psi(\cdot)$ called the *father* and *mother* wavelet, respectively. In greater detail, for any $j_0 \geq 0$, a function $f \in L^2([0, 1])$ can be written as

$$f(x) = \sum_{k=0}^{2^{j_0}-1} \alpha_{j_0 k} \phi_{j_0 k}(x) + \sum_{j=j_0}^{\infty} \sum_{k=0}^{2^j-1} \beta_{jk} \psi_{jk}(x), \tag{1}$$

where

$$\phi_{jk}(x) = 2^{j/2} \phi(2^j x - k), \quad \psi_{jk}(x) = 2^{j/2} \psi(2^j x - k).$$

The coefficients $\alpha_{j_0 k}$ and $\beta_{jk}$ are called the father and mother wavelet coefficients, respectively. The index $j$ is called the *resolution level* and $j_0$ is the *minimum resolution level*. Different choices of $\phi$ and $\psi$ generate various wavelet families. Popular choices are Daubechies [Daubechies, 1988], Coiflets [Daubechies, 1993], Meyer wavelets [Meyer, 1985], and Spline wavelets [Chui, 1992]; for an overview of wavelet families, see Ogden [2012]. Often functions with a truncated basis expansion are considered, i.e., functions of the form $f(x) = \sum_{k=0}^{2^{j_0}-1} \alpha_{j_0 k} \phi_{j_0 k}(x) + \sum_{j=j_0}^{J} \sum_{k=0}^{2^j-1} \beta_{jk} \psi_{jk}(x)$, for some $J$. For regular data with $x_i = i/n$ $(i = 1, \ldots, n)$ and $n = 2^J$ for some $J$, we can calculate the vector $\boldsymbol{f} = [f(1/n), f(2/n), \ldots, f(n/n)]^\top$ efficiently via our second ingredient described next.

Any vector $\boldsymbol{f} = [f(1/n), f(2/n), \ldots, f(n/n)]^\top$, for function $f$ with truncated wavelet basis expansion of order $J$, can be written as a linear combination of that truncated wavelet basis. In particular, $\boldsymbol{f} = W^\top \boldsymbol{d}$, where $\boldsymbol{d} = \left(\alpha_{j_0 0}, \ldots, \alpha_{j_0 2^{j_0}-1}, \beta_{j_0 0}, \beta_{j_0 1}, \ldots, \beta_{J2^J-1}\right)^\top$ is the vector of wavelet coefficients, and the rows of $W$ contain the corresponding wavelet basis functions evaluated at $x_i = i/n$. Specifically, $W$ is an orthogonal matrix with $W_{li} \approx \sqrt{n}\psi_{jk}(i/n)$, or $W_{li} \approx \sqrt{n}\phi_{jk}(i/n)$, for some $l$; the $\sqrt{n}$ factor is due to convention in the literature and software implementation. By orthogonality, $\boldsymbol{d} = W\boldsymbol{f}$; this transformation from $\boldsymbol{f}$ to its wavelet coefficients via multiplication by $W$ is known as the discrete wavelet transform (DWT). The transformation from wavelet coefficients to fitted values, via multiplication by $W^\top$ is known as the inverse discrete wavelet transform (IDWT). The DWT and IDWT can be computed in $O(n)$ operations via Mallat's pyramid algorithm [Mallat, 1989]. However, this is only possible for $n = 2^J$.

Finally, shrinkage is employed to obtain estimates of the form $\widehat{\boldsymbol{f}} = W^\top \widehat{\boldsymbol{d}}$; for ease of exposition, we will assume $j_0 = 0$; i.e., all except the first element of $\boldsymbol{d}$ correspond to mother wavelet coefficients. Our methodology and theoretical results do not depend on the choice of $j_0$. The wavelet shrinkage estimator is given by

$$\widehat{\boldsymbol{d}} \leftarrow \underset{\boldsymbol{d} \in \mathbb{R}^n}{\arg\min} \frac{1}{2}\|\boldsymbol{y} - W^\top \boldsymbol{d}\|_2^2 + \lambda \sum_{i=2}^n |d_i|, \tag{2}$$

for a positive tuning parameter $\lambda$, and given data $\{(i/n, y_i) \in \mathbb{R}^2 : i = 1, \ldots, n\}$. The $\ell_1$ penalty, $\sum_{i=2}^n |d_i| \equiv \|\boldsymbol{d}_{-1}\|_1$, shrinks the wavelet coefficients and also induces sparsity; the sparsity is motivated by the desirable *parsimony* property of wavelets: many functions in $L^2([0,1])$ are sparse linear combinations of wavelet bases. The optimization problem (2) can be solved exactly as follows: define $\widetilde{\boldsymbol{d}} = W\boldsymbol{y}$, the DWT of $\boldsymbol{y}$. Then, $\widehat{d}_1 = \widetilde{d}_1$ and $\widehat{d}_i = sgn(\widetilde{d}_i)(|\widetilde{d}_i| - 2\lambda)_+$ $(i = 2, \ldots, n)$ where $(x)_+ = \max(x, 0)$. Thus, for regularly spaced data with $n = 2^J$, wavelet bases provide an efficient nonparametric estimator. In the following subsection, we discuss some existing methods for dealing with irregularly spaced data and present our novel proposal, `waveMesh`.

## 2.2 A novel interpolation scheme

The common approach to dealing with irregularly spaced data is to map the observed outcomes $\{(x_i, y_i) \in [0,1] \times \mathbb{R} : i = 1, \ldots, n\}$ to approximate outcomes on the regular grid $\{(i/n, y_i') \in \mathbb{R}^2 : i = 1, \ldots, K\}$ for $K = 2^J$ for some integer $J$, via either interpolation or transformation of the data. The novelty of our approach is a reversal of the direction of interpolation, i.e., interpolation from fitted values on the regular grid $i/K$ $(i = 1, \ldots, K)$, to approximated fits on the raw data $x_i$ $(i = 1, \ldots, n)$. For our proposal, we require an interpolation scheme which can be written as a linear map. In greater detail, for any function $f$ evaluated at a regular grid, $\boldsymbol{f} = [f(1/K), \ldots, f(K/K)]^\top$ we require an interpolation scheme $\widetilde{f}(\cdot)$ such that $[\widetilde{f}(x_1), \ldots, \widetilde{f}(x_n)]^\top = R\boldsymbol{f}$ for some interpolation matrix $R \in \mathbb{R}^{n \times K}$. Linear interpolation is a natural choice where

$$\widetilde{f}(x) = f(i/K)\frac{(i+1) - Kx}{(i+1) - i} + f((i+1)/K)\frac{Kx - i}{(i+1) - i}, \tag{3}$$

for $x \in (i/K, (i+1)/K]$ and $\widetilde{f}(x) = f(1/K)$ for $x \leq 1/K$; and the interpolation matrix is

$$R_{ij} = \begin{cases} 1 & j = 1, x_i \leq 1/K \\ (j+1) - Kx_i & j = \lfloor Kx_i \rfloor, x_i \in (1/K, 1] \\ Kx_i - (j-1) & j = \lceil Kx_i \rceil, x_i \in (1/K, 1] \\ 0 & \text{otherwise} \end{cases}. \tag{4}$$

Our proposal, `waveMesh`, solves the following convex optimization problem

$$\widehat{\boldsymbol{d}} \leftarrow \underset{\boldsymbol{d} \in \mathbb{R}^K}{\arg\min} \frac{1}{2}\|\boldsymbol{y} - RW^\top \boldsymbol{d}\|_2^2 + \lambda\|\boldsymbol{d}_{-1}\|_1, \tag{5}$$

where $K = 2^{\lceil \log_2 n \rceil}$, $\boldsymbol{d}_{-1} = [d_2, \ldots, d_n]^\top \in \mathbb{R}^{K-1}$, and $W \in \mathbb{R}^{K \times K}$ is the usual DWT matrix. To evaluate the `waveMesh` estimate at a new point $x \in \mathbb{R}$, one can use $r(x)^\top W^\top \widehat{\boldsymbol{d}}$, where $r$ is given by the chosen interpolation scheme. The advantage of `waveMesh`, over existing methods, is that it can naturally be extended to additive models. Given data $\{(\boldsymbol{x}_i, y_i) \in \mathbb{R}^{p+1} : i = 1, \ldots, n\}$, let $R_j \in \mathbb{R}^{n \times K}$ be the interpolation matrix corresponding to covariate $j$, i.e., $R_j \boldsymbol{f} = [\widetilde{f}(x_{1j}), \ldots, \widetilde{f}(x_{nj})]^\top$. Then, `waveMesh` can be extended to fitting additive models by the following optimization problem:

$$\widehat{\boldsymbol{d}}_1, \ldots, \widehat{\boldsymbol{d}}_p \leftarrow \underset{\boldsymbol{d}_1, \ldots, \boldsymbol{d}_p \in \mathbb{R}^K}{\arg\min} \frac{1}{2}\Big\|\boldsymbol{y} - \sum_{j=1}^p R_j W^\top \boldsymbol{d}_j\Big\|_2^2 + \lambda\sum_{j=1}^p \|\boldsymbol{d}_{j,-1}\|_1, \tag{6}$$

and $\widehat{\boldsymbol{f}} = [\widehat{f}(\boldsymbol{x}_1), \ldots, \widehat{f}(\boldsymbol{x}_n)]^\top = \sum_{j=1}^p \widehat{\boldsymbol{f}}_j = \sum_{j=1}^p R_j W^\top \widehat{\boldsymbol{d}}_j$. Finally, we can extend additive `waveMesh` to fitting sparse additive models for a potentially large number of covariates. This can be achieved by adding a sparsity inducing penalty for each component $f_j$ as follows:

$$\widehat{\boldsymbol{d}}_1, \ldots, \widehat{\boldsymbol{d}}_p \leftarrow \underset{\boldsymbol{d}_1, \ldots, \boldsymbol{d}_p \in \mathbb{R}^K}{\arg\min} \frac{1}{2}\Big\|\boldsymbol{y} - \sum_{j=1}^p R_j W^\top \boldsymbol{d}_j\Big\|_2^2 + \sum_{j=1}^p \big[\lambda_1\|\boldsymbol{d}_{j,-1}\|_1 + \lambda_2\|R_j W^\top \boldsymbol{d}_j\|_2\big]. \tag{7}$$

## 2.3 Algorithm for `waveMesh` and sparse additive `waveMesh`

We now present a proximal gradient descent algorithm [Parikh and Boyd, 2014] for solving the optimization problem (5). For convex loss $\ell$ and penalty $P$, the proximal gradient descent algorithm iteratively finds the minimizer of $\{\ell(\boldsymbol{d}) + P(\boldsymbol{d})\}$ via the iteration:

$$\boldsymbol{d}^{(l+1)} \leftarrow \underset{\boldsymbol{d} \in \mathbb{R}^K}{\arg\min} \frac{1}{2}\Big\|\big(\boldsymbol{d}^{(l)} - t_l \nabla\ell(\boldsymbol{d}^{(l)})\big) - \boldsymbol{d}\Big\|_2^2 + t_l P(\boldsymbol{d}),$$

for a step-size $t_l > 0$. The algorithm is guaranteed to converge as long as $t_l \leq L^{-1}$ where $L$ is the Lipschitz constant of $\nabla\ell(\cdot)$. The step-size can be fixed or selected via a line search algorithm. For (5), we obtain the following iterative scheme:

$$\boldsymbol{d}^{(l+1)} \leftarrow \underset{\boldsymbol{d} \in \mathbb{R}^K}{\arg\min} \frac{1}{2}\Big\|\big\{(I_K - t_l R^\top R)W^\top \boldsymbol{d}^{(l)} + t_l R^\top \boldsymbol{y}\big\} - W^\top \boldsymbol{d}\Big\|_2^2 + t_l \lambda\|\boldsymbol{d}_{-1}^{(l)}\|_1. \tag{8}$$

Our algorithm has a number of desirable features which make it computationally efficient. Firstly, (8) is the traditional wavelet problem for regularly spaced data (2), with response vector $\boldsymbol{r} = \big\{(I_K - t_l R^\top R)W^\top \boldsymbol{d}^{(l)} + t_l R^\top \boldsymbol{y}\big\}$. The vector $\boldsymbol{r}$ can be efficiently calculated via the sparsity of $R$ and Mallat's algorithm for DWT [Mallat, 1989]. Secondly, we can use a fixed step size with $t_l = L_{\max}^{-1}$ where $L_{\max}$ is the maximum eigenvalue of $R^\top R$. Again, the maximum eigenvalue can be efficiently computed for sparse matrices, e.g., if $R$ is the linear interpolation matrix then $R^\top R$ is tridiagonal, and its eigenvalues can be calculated in $O(K \log K)$ operations. The matrix $R$ for linear interpolation matrix needs to be computed once and requires a sorting of the observations, i.e. $O(n \log n)$. Finally, by taking advantage of Nesterov-style acceleration [Nesterov, 2007], the worst-case convergence rate of the algorithm after $k$ steps can be improved from $O(k^{-1})$ to $O(k^{-2})$.

The procedure (8) can also be used to solve the additive (6) and sparse additive (7) extensions via a block coordinate descent algorithm. Specifically, given a set of estimates $\boldsymbol{d}_j$ ($j = 1, \ldots, p$) we can fix all but one of the vectors $\boldsymbol{d}_j$ and optimize over the non-fixed vector, by solving

$$\underset{\boldsymbol{d} \in \mathbb{R}^K}{\text{minimize}} \frac{1}{2}\|\boldsymbol{r}_j - R_j W^\top \boldsymbol{d}\|_2^2 + \lambda_1\|\boldsymbol{d}_{-1}\|_1 + \lambda_2\|R_j W^\top \boldsymbol{d}\|_2, \tag{9}$$

for some vector $\boldsymbol{r}_j \in \mathbb{R}^n$. For additive `waveMesh` ($\lambda_2 = 0$), this reduces to the univariate problem which can be solved via the algorithm (8). For sparse additive `waveMesh` ($\lambda_2 \neq 0$), the problem can be solved by solving (9) with $\lambda_2 = 0$ following by a soft-scaling operation [Petersen et al., 2016, Lemma 7.1]. We detail our algorithm for sparse additive `waveMesh` in the supplementary material.

## 2.4 Some extensions and variations

In this subsection, we discuss some variations and extensions of `waveMesh`, namely (1) using a conservative order for the wavelet basis expansion, (2) extending `waveMesh` for more general loss functions and, (3) using a weighted $\ell_1$ penalty for shrinkage of wavelet coefficients.

While in (5) we set $K = 2^{\lceil \log_2 n \rceil}$, we could, instead, set $K$ to be any power of 2. Since the main computational step in our algorithm is the DWT and IDWT which requires $O(K)$ operations, a smaller value of $K$ can greatly reduce the computation time. Furthermore, using a smaller $K$ can lead to superior predictive performance in some settings; this is formalized in our theoretical results of Section 3 and observed in the simulation studies of Section 4. In the supplementary material we present additional simulation studies comparing the prediction performance and computation time of `waveMesh` for various values of $K$.

Secondly, `waveMesh` can be extended to other loss functions appropriate for various data types. For example, we can extend our methodology to the setting of binary classification via a logistic loss function. Let $y_i \in \{-1, 1\}$ $(i = 1, \ldots, n)$ be the observed response. For the univariate case, we get

$$\widehat{\boldsymbol{d}} \leftarrow \arg\min_{\boldsymbol{d} \in \mathbb{R}^K} \frac{1}{2} \sum_{i=1}^{n} \log\left(1 + \exp\left[-y_i (RW^\top \boldsymbol{d})_i\right]\right) + \lambda \|\boldsymbol{d}_{-1}\|_1. \tag{10}$$

Like the least squares loss, (10) naturally extends to (sparse) additive models. The problem can be efficiently solved via a proximal gradient descent algorithm described in the supplementary material.

Finally, we consider a variation of our $\ell_1$ penalty motivated by the SURESHRINK procedure of Donoho and Johnstone [1995]. For a vector $\boldsymbol{d} \in \mathbb{R}^K$ of discrete father and mother wavelet coefficients, denote by $\boldsymbol{d}_{[j]}$ the discrete mother wavelet coefficients at resolution level $j$. For this particular variation, we require that the minimum resolution level $j_0 > 1$. We then propose to solve

$$\widehat{\boldsymbol{d}} \leftarrow \arg\min_{\boldsymbol{d} \in \mathbb{R}^K} \frac{1}{2} \|\boldsymbol{y} - RW^\top \boldsymbol{d}\|_2^2 + \lambda \sum_{j=j_0}^{\log_2 K} \sqrt{2 \log(j)} \|\boldsymbol{d}_{[j]}\|_1. \tag{11}$$

In the supplementary material we show that the above estimator outperforms the usual `waveMesh` estimator (5) in terms of prediction error.

## 3 Theoretical results

In this section, we study finite sample properties of our univariate estimator (5), and sparse additive estimator (7). We begin with a quick introduction to Besov spaces and their connection to wavelet bases. We establish minimax convergence rates (up to a $\log n$ factor) for our univariate proposal. We note that our estimator (5) can be seen as a lasso estimator [Tibshirani, 1996] with design matrix $RW^\top$; this allows us to use well-known results for the lasso estimator to easily establish minimax rates which we present below. Additionally, the lasso formulation allows us to establish sufficient conditions for the uniqueness of our estimator. Specifically, fitted values $\widehat{\boldsymbol{f}} = RW^\top \widehat{\boldsymbol{d}}$ are unique whereas uniqueness of $\widehat{\boldsymbol{d}}$ depends on the matrix $RW^\top$. In the interest of brevity, we omit derivation of sufficient conditions for uniqueness of $\widehat{\boldsymbol{d}}$ and refer the interested reader to Tibshirani [2013]. Finally, we also establish rates for the sparse additive `waveMesh` proposal for a specific penalty.

Besov spaces on the unit interval, $B^s_{q_1, q_2}$, are function spaces with specific degrees of smoothness in their derivative: for the Besov norm $\| \cdot \|_{B^s_{q_1, q_2}}$, $B^s_{q_1, q_2} = \{g \in L^2([0, 1]) : \|g\|_{B^s_{q_1, q_2}} < C\}$. The constants $(s, q_1, q_2)$ are the parameters of Besov spaces; for a function $g \in L^2([0, 1])$ with the wavelet bases expansion (1), the Besov norm is defined as

$$\|g\|_{B^s_{q_1, q_2}} = \|\boldsymbol{\alpha}_{j_0}\|_{q_1} + \left[ \sum_{j=j_0}^{\infty} \left\{ 2^{j(s+1/2-1/q_1)} \|\boldsymbol{\beta}_j\|_{q_1} \right\}^{q_2} \right]^{1/q_2}, \tag{12}$$

where $\boldsymbol{\alpha}_{j_0} \in \mathbb{R}^{2^{j_0}}$ is the vector of father wavelet coefficients with minimum resolution level $j_0$ and $\boldsymbol{\beta}_j \in \mathbb{R}^{2^j}$ is the vector of mother wavelet coefficients at resolution level $j$. For completeness, we also define $\|g\|_{B^s_{q_1, \infty}} = \|\boldsymbol{\alpha}_{j_0}\|_{q_1} + \sup_{j \geq j_0} \left\{ 2^{j(s+1/2-1/q_1)} \|\boldsymbol{\beta}_j\|_{q_1} \right\}$. We consider Besov spaces

because they generalize well-known classes such as the Sobolev ($B_{2,2}^s$, $s = 1, 2, \ldots$), and Hölder ($B_{\infty,\infty}^s$, $s > 0$) spaces and the class of bounded total variation functions (sandwiched between $B_{1,1}^1$ and $B_{1,\infty}^1$). Our first result below establishes near minimax convergence rates for the prediction error of our estimator. An attractive feature of our estimator is that it achieves this rate without any information about the parameters ($s$, $q_1$, $q_2$). We recover the usual wavelet rates of Donoho [1995] under the special case when $x_i = i/n$ and $R = I_n$. Additionally, the theorem justifies the use of $K < n$ basis functions: if the true function is sufficiently smooth, we recover the usual rates with an additional $\log K$ factor instead of $\log n$.

**Theorem 1** *Suppose $y_i = f^0(x_i) + \varepsilon_i$ $(i = 1, \ldots, n)$ for mean zero, sub-Gaussian noise $\varepsilon_i$. Define the estimator $\widehat{\boldsymbol{f}} = RW^\top \widehat{\boldsymbol{d}} = [\widehat{f}(x_1), \ldots, \widehat{f}(x_n)]^T$ for linear interpolation matrix $R$ (4) where*

$$\widehat{\boldsymbol{d}} \leftarrow \underset{\boldsymbol{d} \in \mathbb{R}^K}{\arg\min} \frac{1}{2}\|\boldsymbol{y} - RW^\top \boldsymbol{d}\|_2^2 + \lambda\|\boldsymbol{d}_{-1}\|_1,$$

*for the usual DWT transform matrix $W \in \mathbb{R}^{K \times K}$ associated with some orthogonal wavelet family. Further, define $\boldsymbol{f}^0 = [f^0(x_1), \ldots, f^0(x_n)]^\top$ and $\widetilde{\boldsymbol{f}}^0 = [f^0(1/K), \ldots, f^0(K/K)]^\top$. Assume that $f^0 \in B_{q_1,q_2}^s$ and the mother wavelet $\psi$, has $r$ null moments and $r$ continuous derivatives where $r > \max\{1, s\}$. Suppose $\lambda \geq c_1\sqrt{t^2 + 2\log K}$ for some $t > 0$. Then, for sufficiently large $K$ (specifically $K \geq c_1 n^{1/(2s+1)}$ for some constant $c_1$), with probability at least $1 - 2\exp(-t^2/2)$, we have*

$$\frac{1}{n}\left\|\boldsymbol{f}^0 - \widehat{\boldsymbol{f}}\right\|_2^2 \leq C\left(\frac{\log K}{n}\right)^{\frac{2s}{2s+1}} + \frac{2}{n}\|\boldsymbol{f}^0 - R\widetilde{\boldsymbol{f}}^0\|_2^2,$$

*where the constant $c_1$ depends on $R$ and the distribution of $\varepsilon_i$, and the constant $C$ depends on $R$.*

The above theorem includes an approximation error term $\|\boldsymbol{f}^0 - R\widetilde{\boldsymbol{f}}^0\|_2^2$ which depends on the type of interpolation matrix $R$. For example, for linear interpolation of a twice continuously differentiable function, the approximation error scales as $O(K^{-2})$. Thus, for a sufficiently large $K$ (particularly $K = n$), the approximation error will disappear. In fact, as long as the approximation error is of the order $(\log K/n)^{2s/(2s+1)}$, we obtain the usual near-minimax rate.

For the sparse additive model, we consider a different model motivated by the Besov norm (12). Our next theorem provides convergence rates for the estimated function $\widehat{\boldsymbol{f}} = \sum_{j=1}^p \widehat{\boldsymbol{f}}_j = \sum_{j=1}^p R_j W^\top \widehat{\boldsymbol{d}}_j$, where

$$\widehat{\boldsymbol{d}}_1, \ldots, \widehat{\boldsymbol{d}}_p \leftarrow \underset{\boldsymbol{d}_1, \ldots, \boldsymbol{d}_p \in \mathbb{R}^K}{\arg\min} \frac{1}{2}\left\|\boldsymbol{y} - \sum_{j=1}^p R_j W^\top \boldsymbol{d}_j\right\|_2^2 + \sum_{j=1}^p \left[\lambda_1 P_s(\boldsymbol{d}_j) + \lambda_2\|R_j W^\top \boldsymbol{d}_j\|_2\right], \quad (13)$$

and the penalty $P_s$ is the discrete version of the Besov norm for $B_{1,1}^s$. Specifically, for $\boldsymbol{d}$ as a vector of father coefficients, $\alpha_{j_0 k}$ $(k = 0, \ldots, 2^{j_0} - 1)$, and mother wavelet coefficients $\beta_{jk}$ $(j = j_0, \ldots, J; k = 0, \ldots, 2^j - 1)$ the penalty is

$$P_s(\boldsymbol{d}) = \sum_{k=0}^{2^{j_0}-1} |\alpha_{j_0 k}| + \sum_{j=j_0}^J \left(2^{j(s-1/2)} \sum_{k=0}^{2^j-1} |\beta_{jk}|\right). \quad (14)$$

Before presenting our next result, we state and discuss the so called *compatibility* condition. This condition is common in the high-dimensional literature [van de Geer and Bühlmann, 2009] and crucial for proving minimax rates for sparse additive models. Briefly, our proof requires the semi-norms $\sum_{j \in S} \|f_j\|_2$ and $\|\sum_{j=1}^p f_j\|_2$ to be somehow 'compatible', for an index set $S \subseteq \{1, \ldots, p\}$. In the low-dimensional/non-sparse case, i.e., $S = \{1, \ldots, p\}$, the semi-norms are compatible by the inequality $\sum_{j \in S} \|f_j\|_2 \leq \sqrt{|S|}\|\sum_{j=1}^p f_j\|_2$. The compatibility condition ensures such an inequality holds for proper subsets $S$. Furthermore, the compatibility condition can be relaxed at the cost of proving a slower rate; this is similar to the lasso *slow rate* [Dalalyan et al., 2017].

**Definition 1** *The compatibility condition is said to hold for an index set $S \subset \{1, 2, \ldots, p\}$, with compatibility constant $\vartheta(S) > 0$, if for all $\gamma > 0$ and any set of discrete wavelet coefficients vector*

$(\boldsymbol{d}_1, \ldots, \boldsymbol{d}_p)$, *that satisfy* $\sum_{j \in S^c} n^{-1} \|R_j W^\top \boldsymbol{d}_j\|_2 + \gamma \sum_{j=1}^p P_s(\boldsymbol{d}_j) \leq 3 \sum_{j \in S} \|R_j W^\top \boldsymbol{d}_j\|$, *it holds that* $\sum_{j \in S} \|R_j W^\top \boldsymbol{d}_j\|_2 \leq \sqrt{|S|} \left\| \sum_{j=1}^p R_j W^\top \boldsymbol{d}_j \right\|_2 / \vartheta(S)$.

**Theorem 2** *Assume the model* $y_i = f^0(\boldsymbol{x}_i) + \varepsilon_i$ $(i = 1, \ldots, n)$ *with mean zero, sub-Gaussian* $\varepsilon_i$. *Let* $\widehat{f} = \sum_{j=1}^p \widehat{\boldsymbol{f}}_j$ *be as defined in* (13), *and let* $\boldsymbol{f}^* = \sum_{j \in S^*} \boldsymbol{f}_j^* = \sum_{j \in S^*} R_j W^\top \boldsymbol{d}_j^*$ *be an arbitrary sparse additive function with* $S^* \subset \{1, 2, \ldots, p\}$. *Let* $\rho = \kappa \max\{n^{-2s/(2s+1)}, (\log p/n)^{1/2}\}$ *for a constant* $\kappa$ *that depends on the distribution of* $\varepsilon_i$ *and* $s$. *Suppose* $\lambda \geq 4\rho$. *Then, with probability at least* $1 - 2\exp(-c_1 n\rho^2) - c_2 \exp(-c_3 n\rho^2)$, *we have*

$$n^{-1} \left\| \boldsymbol{f}^0 - \widehat{\boldsymbol{f}} \right\|_2^2 \leq C_1 \max \left\{ |S^*| n^{-\frac{s}{2s+1}}, |S^*| \left( \frac{\log p}{n} \right)^{1/2} \right\} + n^{-1} \left\| \boldsymbol{f}^0 - \boldsymbol{f}^* \right\|_2^2,$$

*where constants* $c_1, c_2$ *depend on the distribution of* $\varepsilon_i$ *and* $s$, *and* $C_1$ *depends on* $\kappa$ *and* $|S^*|^{-1} \sum_{j \in S^*} P_s(\boldsymbol{d}_j^*)$. *Furthermore, if the compatibility condition holds for* $S^*$ *with constant* $\vartheta(S^*)$ *we have*

$$n^{-1} \left\| \boldsymbol{f}^0 - \widehat{\boldsymbol{f}} \right\|_2^2 \leq C_2 \max \left\{ |S^*| n^{-\frac{2s}{2s+1}}, |S^*| \frac{\log p}{n} \right\} + 4n^{-1} \left\| \boldsymbol{f}^0 - \boldsymbol{f}^* \right\|_2^2,$$

*where the constant* $C_2$ *depends on* $\vartheta(S^*)$ *and* $|S^*|^{-1} \sum_{j \in S^*} P_s(\boldsymbol{d}_j^*)$.

## 4 Numerical experiments

### 4.1 Experiments for univariate regression

We begin with a simulation to compare the performance of univariate `waveMesh` to the traditional interpolation method of Kovac and Silverman [2000], isometric wavelet method of Sardy et al. [1999]—which treats the data as if it were regularly spaced—and adaptive lifting method of Nunes et al. [2006]. The former two methods are implemented in the `R` package `wavethres` [Nason, 2016] and the latter is implemented in the `adlift` package [Nunes and Knight, 2017].

We generate the data as $y_i = f^0(x_i) + \varepsilon_i$ $(i = 1, \ldots, n)$ for different choices of function $f^0$ and $n$. The errors are distributed as $\varepsilon_i \sim \mathcal{N}(0, \sigma^2)$ with $\sigma^2$ chosen such that SNR = 5, where SNR = $\text{var}(\boldsymbol{f}^0)/\sigma^2$. We consider two different choices of the covariate, $x_i \sim \mathcal{U}[0,1]$ and $x_i \sim \mathcal{N}(0,1)$ scaled to lie in $[0,1]$. We consider 6 different choices for the function $f^0$: 1. polynomial, 2. sine, 3. piecewise polynomial, 4. heavy sine, 5. bumps and, 6. doppler. These functions are shown in Figure 1 of the supplementary material. We apply our proposal, `waveMesh`, the interpolation proposal of Kovac and Silverman [2000] and isometric wavelet proposal of Sardy et al. [1999], for a sequence of 50 $\lambda$ values linear on the log scale and select the $\lambda$ value that minimizes the mean square error, MSE $= n^{-1} \|\boldsymbol{f}^0 - \widehat{\boldsymbol{f}}\|_2^2$. For adaptive lifting, the `R` implementation automatically selects a tuning parameter. We implement `waveMesh` using the linear interpolation matrix (4). We also implement `waveMesh` using a small grid, i.e., we fit (5) with $K = 2^5$ and $2^6$. The `R` implementation of isometric wavelets requires sample sizes to be a power of two; if not, we pad the response vector with zeros.

We also analyze the motorcycle data studied by Silverman [1985] consisting of 133 head acceleration measurements in a simulated motorcycle accident taken at 94 unequally spaced time points. To avoid the issue of repeated measurements, we average acceleration measurements at the same time leading to a sample size of $n = 94$. Selection of tuning parameter for `waveMesh` is done via 5-fold cross validation. For interpolation [Sardy et al., 1999] and isometric [Kovac and Silverman, 2000] wavelet proposals, we use the *universal thresholding* rule for tuning parameter selection [Donoho and Johnstone, 1994]; this rule leads to near minimax convergence rates like that of Theorem 1.

Table 1 shows the ratio of MSE between our proposal with $K = 2^{\lceil \log_2 n \rceil}$ and other proposals for uniformly distributed $x_i$. We observe that our proposal has the smallest MSE for all functions except the Bumps function. Even for the Bumps function, `waveMesh` exhibits superior prediction performance over other methods for $n = 512$. We also observe that `waveMesh` with smaller values of $K$ often outperforms the full `waveMesh` ($K = 2^{\lceil \log_2 n \rceil}$) method in terms of MSE. Results for normally distributed $x_i$ are given in the supplementary material. In that case, we again observe that `waveMesh` outperforms existing methods for a number of simulation scenarios, except for a few cases with polynomial and bumps functions. Results for sample sizes that are not powers of two

Table 1: Results for $x_i \sim \mathcal{U}[0,1]$ averaged over 100 replicates; the ratio $\text{MSE}/\text{MSE}_{FG}$ is shown along with $100\times$ the standard error, where $\text{MSE}_{FG}$ is the MSE of `waveMesh` with $K = 2^{\lceil \log_2 n \rceil}$. Boldface values represent the method with the smallest MSE within each row of the table.

| | | waveMesh $K = 2^5$ | waveMesh $K = 2^6$ | Interpolation | Isometric | Adaptive Lifting |
|---|---|---|---|---|---|---|
| Polynomial | n = 64 | 1.19 (5.51) | **1.00** (0.00) | 1.24 (4.11) | 1.78 (7.56) | 4.28 (29.86) |
| | n = 128 | 0.92 (5.57) | **0.77** (3.07) | 1.12 (6.00) | 1.33 (7.18) | 3.57 (31.27) |
| | n = 256 | 1.00 (6.20) | **0.85** (3.15) | 1.61 (9.04) | 1.50 (7.67) | 4.29 (31.29) |
| | n = 512 | 0.78 (3.18) | **0.72** (2.58) | 1.76 (6.11) | 1.13 (2.64) | 3.61 (26.47) |
| Sine | n = 64 | **0.97** (3.14) | 1.00 (0.00) | 1.47 (5.81) | 1.59 (6.72) | 3.62 (33.65) |
| | n = 128 | **0.76** (3.18) | 0.76 (1.96) | 1.29 (6.08) | 1.46 (5.24) | 2.98 (19.78) |
| | n = 256 | **0.66** (2.50) | 0.70 (2.22) | 1.93 (9.49) | 1.34 (4.23) | 3.41 (18.80) |
| | n = 512 | 0.57 (2.34) | **0.56** (2.22) | 2.13 (7.78) | 1.24 (3.66) | 3.63 (28.42) |
| Piecewise | n = 64 | **0.85** (1.97) | 1.00 (0.00) | 1.18 (3.12) | 1.31 (3.62) | 1.63 (9.07) |
| Polynomial | n = 128 | **0.77** (2.00) | 0.82 (1.52) | 1.26 (2.75) | 1.22 (2.61) | 1.40 (7.36) |
| | n = 256 | 0.82 (1.92) | **0.79** (1.59) | 1.42 (3.18) | 1.14 (2.11) | 1.15 (6.04) |
| | n = 512 | 1.01 (2.43) | **0.86** (1.70) | 1.71 (3.56) | 1.15 (1.99) | 1.25 (7.24) |
| Heavy Sine | n = 64 | **0.84** (2.44) | 1.00 (0.00) | 1.12 (3.04) | 1.41 (3.17) | 1.70 (8.35) |
| | n = 128 | **0.75** (2.66) | 0.82 (1.16) | 1.17 (3.32) | 1.50 (4.75) | 1.56 (8.26) |
| | n = 256 | **0.66** (1.64) | 0.72 (1.14) | 1.37 (2.98) | 1.33 (2.58) | 1.53 (6.74) |
| | n = 512 | **0.58** (1.59) | 0.60 (1.18) | 1.58 (3.05) | 1.29 (1.60) | 1.50 (9.21) |
| Bumps | n = 64 | 2.11 (2.30) | 1.00 (0.00) | 1.70 (1.75) | **0.72** (1.34) | 1.07 (5.12) |
| | n = 128 | 2.86 (2.77) | 2.11 (1.62) | 1.40 (1.59) | **0.63** (0.83) | 0.85 (2.43) |
| | n = 256 | 4.81 (6.82) | 3.47 (4.39) | 1.43 (1.89) | **0.88** (0.99) | 0.97 (2.00) |
| | n = 512 | 7.45 (9.13) | 5.69 (6.77) | 1.32 (1.35) | **1.19** (1.03) | 1.23 (2.34) |
| Doppler | n = 64 | **0.98** (1.69) | 1.00 (0.00) | 1.15 (3.45) | 1.33 (3.20) | 1.30 (3.65) |
| | n = 128 | 1.24 (2.02) | **0.89** (1.04) | 1.07 (2.13) | 1.44 (2.57) | 1.18 (3.22) |
| | n = 256 | 1.71 (3.92) | **0.94** (1.38) | 1.20 (2.11) | 1.29 (1.99) | 1.30 (3.44) |
| | n = 512 | 2.58 (4.85) | 1.26 (2.01) | 1.21 (1.48) | **1.10** (1.31) | 1.23 (3.36) |

were similar to the results provided here. In the interest of brevity, these results are presented in the supplementary material.

In Figure 1, we plot the motorcycle data and fitted functions for each method. Here, `waveMesh` reasonably models the data via a smooth function; the interpolation method has a similar but slightly more biased result around 10 to 25 ms. Adaptive lifting and isometric wavelets lead to highly variable estimates.

## 4.2  Experiments for multivariate additive regression

We proceed with a simulation study to illustrate the performance of additive `waveMesh` compared to the proposal of Sardy and Tseng [2004], AMlet. We use the author-provided R implementation for the AMlet proposal; due to a lack of R packages for other proposals, we defer the comparison to future work. We consider the following simulation setting: we generate data with $y_i = f_1(x_{i1}) + f_2(x_{i2}) + f_3(x_{i3}) + f_4(x_{i4}) + \varepsilon_i$ ($i = 1, \ldots, 2^{10}$), where $\varepsilon_i \sim \mathcal{N}(0, \sigma^2)$, $x_i \sim \mathcal{U}[0,1]$, and $\sigma^2$ such that SNR = 10. The four functions $f_1, \ldots, f_4$ are the polynomial, sine, piecewise polynomial and heavy sine functions presented in Figure 1 of the supplementary material. We consider sample sizes $n = 64, 100, 256, 500, 512$ and results were averaged over 100 data sets. For sample sizes not a power of 2, the response vector was padded with zeros for the R implementation of AMlet. The universal threshold rule was used for AMlet as detailed in Sardy and Tseng [2004]; 5-fold cross validation was used for additive `waveMesh` for selection of $\lambda$.

For a real world data analysis, we consider the Boston housing data analyzed by Ravikumar et al. [2009]. The goal is to predict the median value of homes based on 10 predictors. The data consists of $n = 506$ observations; we use 256 observations for training and calculate the test error on the rest. Tuning parameters are selected in the same way as the simulation study above.

Table 2 shows the MSE of both proposals for various choices of $n$ for the simulation study. The results clearly indicate that additive `waveMesh` offers substantial improvement over AMlet, especially

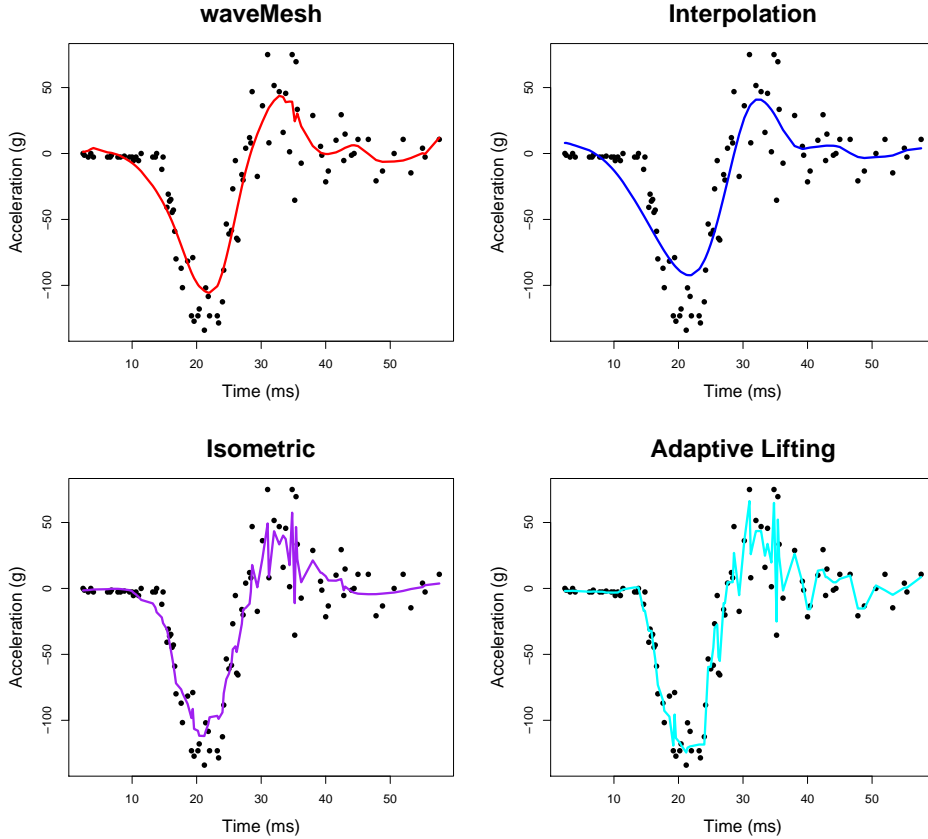

Figure 1: Fitted functions to the motorcycle accident dataset for each of the 4 methods.

for smaller values of $n$. We observe similar results for the Boston housing data: the average test error is 21.2 for `waveMesh` (standard error 0.34) and 25.1 for AMlet (standard error 0.42). These results support our theoretical analysis and underscore the advantages of `waveMesh` in sparse high-dimensional additive models.

## 5   Conclusion

In this paper, we introduced `waveMesh`, a novel method for non-parametric regression using wavelets. Unlike traditional methods, `waveMesh` does not require that covariates are uniformly spaced on the unit interval, nor does it require that the sample size is a power of 2. We achieve this using a novel interpolation approach for wavelets. The main appeal of our proposal is that it naturally extends to multivariate additive models for a potentially large number of covariates.

To compute the estimator, we proposed an efficient proximal gradient descent algorithm, which leverages existing techniques for fast computation of the DWT. We established minimax convergence rates for our univariate proposal over a large class of Besov spaces. For a particular Besov space, we also established minimax convergence rates for our (sparse) additive framework. The `R` package `waveMesh`, which implements our methodology, will soon be publicly available on GitHub.

Table 2: MSE and standard error of `waveMesh` and AMlet averaged over 100 data sets.

|  | $n = 64$ | $n = 100$ | $n = 128$ | $n = 256$ | $n = 500$ | $n = 512$ |
|---|---|---|---|---|---|---|
| `waveMesh` | **10.76** (0.31) | **11.35** (0.33) | **8.82** (0.24) | **5.45** (0.11) | **4.34** (0.08) | **4.08** (0.07) |
| AMlet | 100.48 (1.83) | 34.58 (1.05) | 45.49 (1.09) | 19.57 (0.33) | 10.67 (0.12) | 8.90 (0.11) |

**Acknowledgments**

We thank three anonymous referees for insightful comments that substantially improved the manuscript. We thank Professor Sylvain Sardy for providing software. This work was partially supported by National Institutes of Health grants to A.S. and N.S., and National Science Foundation grants to A.S.

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
