[Supplementary Material]

# Supplementary material for wavelet regression and additive models for irregularly spaced data

**Asad Haris**[*]
Department of Biostatistics
University of Washington
Seattle, WA 98195
aharis@uw.edu

**Noah Simon**
Department of Biostatistics
University of Washington
Seattle, WA 98195
nrsimon@uw.edu

**Ali Shojaie**
Department of Biostatistics
University of Washington
Seattle, WA 98195
ashojaie@uw.edu

## 1 Details of Algorithms

Here we give an algorithm for our additive and sparse-additive framework as well as an algorithm for the extension of our proposal to classification. We use a block-wise coordinate descent algorithm for solving the additive and sparse additive proposal. This algorithm cyclically iterates through features, and for each feature applies the univariate solution detailed in the main manuscript. The exact details are given in Algorithm 1 below.

---

Initialize $\boldsymbol{d}_j \leftarrow 0$ for $j = 1, \ldots, p$
While $l \leq max\_iter$ **and** not converged
    For $j = 1, \ldots, p$
        Set $\boldsymbol{r}_{-j} \leftarrow \boldsymbol{y} - \sum_{j' \neq j} R_{j'} W^\top \boldsymbol{d}_{j'}$
        Update $\boldsymbol{d}_j \leftarrow \underset{\boldsymbol{d} \in \mathbb{R}^K}{\arg\min} \frac{1}{2} \left\| r_{-j} - R_j W^\top \boldsymbol{d} \right\|_2^2 + \lambda_1 \|\boldsymbol{d}_{-1}\|_1 + \lambda_2 \|R_j W^\top \boldsymbol{d}\|_2,$
Return $\boldsymbol{d}_1, \ldots, \boldsymbol{d}_p$

**Algorithm 1:** Block coordinate descent for the additive and sparse additive framework

---

We also give an algorithm for the extension of our method to classification based on proximal gradient descent. To begin let $L(\boldsymbol{d}) = 1/(2n) \sum_{i=1}^n \log \left(1 + \exp \left[ -y_i \left\{ \left( RW^\top \boldsymbol{d} \right)_i \right\} \right]\right)$, or more generally let it be some differentiable convex loss function. We denote by $\nabla L(\boldsymbol{d})$, the derivative of $L$ at the point $\boldsymbol{d} \in \mathbb{R}^K$. Algorithm 2 presents the steps for solving the univariate waveMesh problem with general loss. The algorithm for extension of additive models to classification (or other loss functions) can be similarly derived and is omitted in the interest of brevity.

## 2 Additional simulation results

In this section we present some additional simulation results. The simulation study for both univariate and multivariate regression, used six functions: 1. polynomial, 2. sine, 3. piecewise polynomial, 4. heavy sine, 5. bumps and, 6. doppler. The six functions are presented in Figure 1.

---

[*]Mailing address: Box 357232, University of Washington, Seattle, WA 98195-7232

Initialize $\boldsymbol{d}^0$

For $l = 1, 2, \ldots$ until convergence
      Select a step size $t_l$ via line search
      Update

$$\boldsymbol{d}^l \leftarrow \underset{\boldsymbol{d} \in \mathbb{R}^K}{\arg\min} \frac{1}{2} \left\| \boldsymbol{d} - \left\{ \boldsymbol{d}^{l-1} - t_l \nabla L(\boldsymbol{d}^{l-1}) \right\} \right\|_2^2 + t_l \lambda \|\boldsymbol{d}_{-1}\|_1.$$

Return $\boldsymbol{d}^l$

**Algorithm 2:** Proximal gradient descent for extension to classification

Figure 1: Plots of functions $f^0$ for the simulation study. Functions in green are the most smooth and well-behaved followed by functions with moderate smoothness in orange. Finally, functions in red are highly irregular functions, e.g., functions with unbounded total variation.

## 2.1 Univariate simulation study for $x_i \sim \mathcal{N}(0,1)$

We begin with presenting the table of results for the univariate regression simulation study. In Table 1, we present the results for normally distributed covariates, i.e., $x_i \sim \mathcal{N}(0,1)$, and then scaled to $[0,1]$. We see that other than the polynomial function `waveMesh` generally outperforms competitors in terms of prediction error.

## 2.2 Univariate simulation study for sample sizes not a power of two

In this section, we present results for the simulation study of Section 4 for sample sizes $n = 75, 100, 300, 500$. The results are presented for $x_i \sim \mathcal{U}(0,1)$ and $x_i \sim \mathcal{N}(0,1)$ in Table 2 and 3, respectively.

## 2.3 Effect of truncation level $K$

In this subsection, we present simulation results which study the effects of using different truncation levels $K$. In Figures 2 to 7 we plot the results for each of the 6 functions considered in the simulation of the manuscript.

Table 1: Table of results for $x_i \sim \mathcal{N}(0,1)$ averaged over 100 replications of the data. The table presents the ratio MSE / $\text{MSE}_{FG}$ along with $100\times$ the standard error, where $\text{MSE}_{FG}$ is the MSE of `waveMesh` with $K = 2^{\lceil \log_2 n \rceil}$. Boldface values represent the method with the smallest MSE within each row of the table.

| | | waveMesh $K = 2^5$ | waveMesh $K = 2^6$ | Interpolation | Isometric | Adaptive Lifting |
|---|---|---|---|---|---|---|
| Polynomial | n = 64 | 1.47 (13.17) | 1.41 (11.32) | **0.51** (3.04) | 1.45 (10.11) | 1.59 (9.30) |
| | n = 128 | 0.78 (5.25) | 0.77 (4.95) | **0.40** (2.96) | 0.87 (4.69) | 0.88 (4.84) |
| | n = 256 | **0.39** (3.75) | 0.51 (3.89) | 0.43 (2.38) | 0.64 (2.81) | 0.76 (5.97) |
| | n = 512 | 0.90 (4.57) | 0.77 (4.03) | **0.29** (0.98) | 0.43 (1.59) | 0.33 (2.36) |
| Sine | n = 64 | 0.92 (9.55) | 0.99 (1.49) | 1.48 (11.85) | 2.22 (21.67) | 3.61 (35.07) |
| | n = 128 | **0.89** (8.74) | 0.91 (3.77) | 1.71 (10.85) | 1.83 (15.18) | 3.53 (33.07) |
| | n = 256 | **0.48** (2.39) | 0.73 (1.53) | 1.48 (8.74) | 1.51 (8.18) | 2.73 (22.25) |
| | n = 512 | **0.36** (1.22) | 0.64 (1.63) | 1.03 (5.54) | 0.74 (2.77) | 1.21 (7.62) |
| Piecewise | n = 64 | **0.78** (1.92) | 0.99 (1.01) | 1.50 (6.50) | 1.64 (7.54) | 2.18 (14.06) |
| Polynomial | n = 128 | 0.86 (2.29) | **0.83** (2.04) | 1.89 (7.42) | 1.59 (4.60) | 1.65 (8.86) |
| | n = 256 | 1.25 (3.80) | **0.90** (2.22) | 1.64 (5.21) | 1.09 (3.24) | 1.15 (6.94) |
| | n = 512 | 1.79 (2.71) | 1.27 (2.34) | 1.76 (3.24) | **0.96** (1.54) | 1.01 (4.29) |
| Heavy Sine | n = 64 | **0.73** (1.81) | 1.00 (0.65) | 1.23 (4.40) | 1.26 (4.03) | 1.54 (6.83) |
| | n = 128 | **0.54** (1.70) | 0.78 (1.40) | 1.30 (5.02) | 1.14 (2.78) | 1.12 (6.04) |
| | n = 256 | **0.47** (0.93) | 0.65 (0.98) | 1.17 (3.08) | 0.89 (1.99) | 0.93 (5.45) |
| | n = 512 | **0.38** (0.87) | 0.54 (1.08) | 1.40 (2.91) | 0.77 (1.24) | 0.84 (3.94) |
| Bumps | n = 64 | 1.27 (0.62) | 1.00 (0.06) | 0.85 (1.19) | **0.36** (0.79) | 0.53 (2.24) |
| | n = 128 | 3.40 (4.69) | 2.25 (2.81) | 1.35 (2.28) | **0.69** (1.50) | 0.76 (1.64) |
| | n = 256 | 6.49 (10.88) | 3.71 (5.58) | 1.31 (2.03) | 1.18 (1.41) | **1.10** (2.52) |
| | n = 512 | 8.83 (10.06) | 5.43 (6.03) | 1.29 (1.82) | 1.28 (1.37) | **1.11** (1.90) |
| Doppler | n = 64 | **0.75** (1.84) | 1.00 (0.67) | 1.36 (4.74) | 1.53 (4.32) | 1.56 (6.01) |
| | n = 128 | 0.99 (1.87) | **0.81** (1.44) | 1.43 (4.75) | 1.49 (3.81) | 1.40 (4.35) |
| | n = 256 | 0.58 (1.11) | **0.52** (1.06) | 1.26 (3.25) | 1.15 (1.86) | 0.98 (3.77) |
| | n = 512 | 0.98 (1.52) | **0.58** (1.05) | 1.24 (2.38) | 0.98 (1.48) | 0.85 (2.21) |

In the left panel of each figure we plot the MSE as a function of sample size, $n$. This is done for the full grid method where we take $K = 2^{\log_2 n}$, and for `waveMesh` with $K = 2^4, 2^5$ and $2^6$ which we refer to as 4 Grid, 5 Grid and 6 Grid, respectively. In the right panel of each figure we present the computation time as a function of sample size $n$ for `waveMesh` with $K = 2^4, 2^5, 2^6$ and $2^{\log_2 n}$.

We see in Figures 6 and 7, that using a small order $K$ leads to substantially high MSE. This is most likely due to the nature of the underlying functions. The Doppler function is an example of function which does not have a bounded variation, estimating such functions by interpolation is extremely difficult and in general we need a full grid, i.e. $K = n$. On the other hand for all other functions, i.e. polynomial, sine etc, we see a clear advantage of using $K = 2^7$ basis functions. We also see in some figures that while using $K = 2^6$ leads to substantially smaller MSE using too small a value of $K$ can be lead to poor prediction performance. We see this even in the simple cases of estimating a polynomial or sine function.

We notice on the right panels the clear computational advantage of using fewer than $n$ basis functions. We observe the computation time for fixed $K$ generally does not vary too much with increasing sample size. This is because the main computational step is the DWT and IDWT via Mallets algorithm. The other matrix multiplications are sparse and can be computed efficiently.

## 2.4 Simulation study for adaptive `waveMesh`

Finally, in this subsection, we present some simulation results regarding the adaptive `waveMesh` estimator introduced in Section 2.4 of the Manuscript. In the left panel of Figure 8 to 13 we present the MSE as a function of sample size for regular `waveMesh` with $K = n$ and adaptive `waveMesh`. We present the minimum MSE over a sequence of 50 $\lambda$ values. We see that our adaptive estimator uniformly outperforms the regular estimator in terms of prediction error. The results indicates that if

Table 2: Table of results for $x_i \sim \mathcal{U}[0,1]$ averaged over 100 replications of the data for sample sizes that are not powers of 2. The table presents the ratio $\mathrm{MSE}/\mathrm{MSE}_{FG}$ along with $100\times$ the standard error, where $\mathrm{MSE}_{FG}$ is the MSE of `waveMesh` with $K = 2^{\lceil \log_2 n \rceil}$. Boldface values represent the method with the smallest MSE within each row of the table.

| | | waveMesh $K = 2^5$ | waveMesh $K = 2^6$ | Interpolation | Isometric | Adaptive Lifting |
|---|---|---|---|---|---|---|
| Polynomial | n = 75 | 1.20 (3.51) | **1.00** (0.00) | 1.32 (3.71) | 4.98 (22.10) | 4.35 (24.12) |
| | n = 100 | 1.18 (3.96) | **1.00** (0.00) | 1.39 (4.76) | 4.24 (17.06) | 3.98 (21.99) |
| | n = 300 | 0.84 (3.02) | **0.81** (2.55) | 1.87 (5.91) | 5.33 (18.57) | 3.86 (19.36) |
| | n = 500 | 0.96 (2.95) | **0.89** (2.69) | 2.13 (5.91) | 3.36 (14.51) | 4.21 (21.17) |
| Sine | n = 75 | 1.09 (3.76) | **1.00** (0.00) | 1.55 (5.52) | 2.57 (11.78) | 3.78 (21.66) |
| | n = 100 | 1.04 (2.73) | **1.00** (0.00) | 1.67 (6.81) | 1.75 (5.65) | 3.43 (19.26) |
| | n = 300 | **0.67** (1.77) | 0.73 (1.95) | 2.33 (7.06) | 2.18 (6.65) | 4.26 (25.07) |
| | n = 500 | **0.73** (2.08) | 0.76 (2.42) | 2.72 (8.77) | 1.28 (3.35) | 4.05 (21.45) |
| Piecewise | n = 75 | **0.87** (1.55) | 1.00 (0.00) | 1.32 (2.60) | 1.40 (3.16) | 1.73 (7.43) |
| Polynomial | n = 100 | **0.84** (1.42) | 1.00 (0.00) | 1.33 (2.93) | 1.39 (2.59) | 1.40 (5.63) |
| | n = 300 | 0.98 (1.47) | **0.92** (1.25) | 1.63 (2.51) | 1.27 (1.68) | 1.40 (5.28) |
| | n = 500 | 1.19 (1.57) | **1.03** (1.13) | 1.95 (3.23) | 1.26 (1.48) | 1.36 (4.01) |
| Heavy Sine | n = 75 | **0.89** (1.87) | 1.00 (0.00) | 1.31 (2.92) | 1.44 (3.01) | 1.79 (6.43) |
| | n = 100 | **0.87** (1.49) | 1.00 (0.00) | 1.38 (2.92) | 1.72 (3.50) | 1.59 (5.33) |
| | n = 300 | **0.73** (1.39) | 0.81 (1.01) | 1.87 (3.03) | 1.87 (2.93) | 1.80 (5.81) |
| | n = 500 | **0.76** (1.23) | 0.80 (1.04) | 1.99 (3.17) | 1.61 (1.97) | 1.77 (5.37) |
| Bumps | n = 75 | 1.83 (1.10) | 1.00 (0.00) | 0.76 (1.00) | **0.46** (0.61) | 0.88 (3.90) |
| | n = 100 | 1.56 (0.59) | 1.00 (0.00) | 0.72 (0.80) | **0.38** (0.48) | 0.61 (1.75) |
| | n = 300 | 4.47 (3.00) | 3.20 (1.99) | 0.87 (0.54) | **0.81** (0.60) | 0.83 (1.17) |
| | n = 500 | 4.57 (2.03) | 3.51 (1.48) | 0.80 (0.52) | **0.74** (0.53) | 0.74 (0.73) |
| Doppler | n = 75 | **0.96** (1.39) | 1.00 (0.00) | 1.19 (1.95) | 1.47 (2.54) | 1.40 (3.54) |
| | n = 100 | 1.18 (1.30) | **1.00** (0.00) | 1.25 (2.13) | 1.50 (2.33) | 1.37 (2.97) |
| | n = 300 | 2.27 (3.46) | **1.10** (1.15) | 1.36 (1.53) | 1.36 (1.51) | 1.37 (2.30) |
| | n = 500 | 3.44 (4.69) | 1.70 (1.98) | 1.60 (1.69) | **1.42** (1.60) | 1.59 (2.29) |

Figure 2: Effect of truncation level $K$. Results of for the Polynomial function.

Table 3: Table of results for $x_i \sim \mathcal{N}(0,1)$ averaged over 100 replications of the data for sample sizes that are not powers of 2. The table presents the ratio $\mathrm{MSE} / \mathrm{MSE}_{FG}$ along with $100\times$ the standard error, where $\mathrm{MSE}_{FG}$ is the MSE of `waveMesh` with $K = 2^{\lceil \log_2 n \rceil}$. Boldface values represent the method with the smallest MSE within each row of the table.

| | | waveMesh $K = 2^5$ | waveMesh $K = 2^6$ | Interpolation | Isometric | Adaptive Lifting |
|---|---|---|---|---|---|---|
| Polynomial | $n = 75$ | 1.11 (3.83) | 1.00 (0.00) | **0.99** (4.31) | 3.93 (18.03) | 3.10 (21.66) |
| | $n = 100$ | 1.26 (4.25) | **1.00** (0.00) | 1.16 (4.45) | 4.24 (21.09) | 2.94 (17.20) |
| | $n = 300$ | 0.68 (2.77) | **0.49** (1.11) | 1.01 (3.27) | 3.19 (7.76) | 1.74 (8.14) |
| | $n = 500$ | 0.37 (0.93) | **0.37** (0.77) | 0.80 (2.01) | 1.57 (2.39) | 0.92 (4.36) |
| Sine | $n = 75$ | **0.86** (2.86) | 1.00 (0.00) | 1.16 (5.27) | 1.81 (6.86) | 2.64 (12.89) |
| | $n = 100$ | **0.87** (2.66) | 1.00 (0.00) | 1.24 (5.00) | 1.82 (6.40) | 2.49 (13.27) |
| | $n = 300$ | **0.67** (1.75) | 0.86 (1.53) | 1.12 (4.43) | 1.63 (5.41) | 1.77 (8.27) |
| | $n = 500$ | 0.74 (2.10) | **0.73** (1.50) | 1.41 (4.43) | 1.23 (3.16) | 1.69 (7.46) |
| Piecewise Polynomial | $n = 75$ | **0.95** (1.90) | 1.00 (0.00) | 1.54 (4.27) | 1.65 (4.39) | 1.74 (7.56) |
| | $n = 100$ | **0.96** (1.95) | 1.00 (0.00) | 1.70 (4.76) | 1.54 (4.35) | 1.55 (6.54) |
| | $n = 300$ | 1.32 (2.34) | **0.91** (1.07) | 1.76 (3.67) | 1.27 (2.43) | 1.25 (4.37) |
| | $n = 500$ | 1.76 (2.45) | 1.20 (1.41) | 1.79 (3.23) | **0.95** (1.54) | 1.03 (3.29) |
| Heavy Sine | $n = 75$ | **0.81** (1.47) | 1.00 (0.00) | 1.25 (3.14) | 1.48 (2.68) | 1.56 (5.69) |
| | $n = 100$ | **0.85** (2.11) | 1.00 (0.00) | 1.48 (4.03) | 1.68 (3.34) | 1.47 (5.58) |
| | $n = 300$ | **0.55** (1.34) | 0.66 (1.14) | 1.33 (2.59) | 1.47 (1.93) | 1.01 (3.28) |
| | $n = 500$ | 0.73 (1.85) | **0.72** (1.12) | 1.71 (2.80) | 1.12 (1.47) | 1.03 (3.56) |
| Bumps | $n = 75$ | 1.28 (0.43) | 1.00 (0.00) | 0.69 (0.59) | **0.34** (0.54) | 0.48 (1.66) |
| | $n = 100$ | 1.42 (0.52) | 1.00 (0.00) | 0.65 (0.56) | **0.35** (0.49) | 0.41 (1.34) |
| | $n = 300$ | 4.58 (3.42) | 3.04 (2.07) | 0.83 (0.84) | 0.83 (0.76) | **0.81** (0.98) |
| | $n = 500$ | 7.20 (5.39) | 4.46 (3.24) | 1.08 (1.03) | 1.10 (1.01) | **0.91** (1.11) |
| Doppler | $n = 75$ | 1.09 (1.70) | **1.00** (0.00) | 1.37 (3.22) | 1.63 (3.08) | 1.58 (3.96) |
| | $n = 100$ | 1.23 (1.69) | **1.00** (0.00) | 1.46 (3.17) | 1.77 (3.39) | 1.61 (4.31) |
| | $n = 300$ | 0.68 (1.29) | **0.66** (1.11) | 1.67 (2.59) | 1.51 (2.34) | 1.21 (2.99) |
| | $n = 500$ | 1.36 (1.69) | **0.82** (0.77) | 1.79 (2.63) | 1.35 (1.78) | 1.22 (2.61) |

Figure 3: Effect of truncation level $K$. Results of for the Sine function.

Figure 4: Effect of truncation level $K$. Results of for the Piecewise Polynomial function.

Figure 5: Effect of truncation level $K$. Results of for the Heavy sine function.

Figure 6: Effect of truncation level $K$. Results of for the Doppler function.

Figure 7: Effect of truncation level $K$. Results of for the Bumps function.

Figure 8: Simulation study for adaptive `waveMesh`. Results of for the Polynomial function.

Figure 9: Simulation study for adaptive `waveMesh`. Results of for the Sine function.

we have a good procedure for selecting the tuning parameter, i.e., if we pick close to the theoretically ideal tuning parameter then adaptive `waveMesh` will have a lower MSE.

Figure 10: Simulation study for adaptive `waveMesh`. Results of for the Piecewise Polynomial function.

Figure 11: Simulation study for adaptive `waveMesh`. Results of for the Heavy sine function.

Figure 12: Simulation study for adaptive `waveMesh`. Results of for the Doppler function.

Figure 13: Simulation study for adaptive `waveMesh`. Results of for the Bumps function.

# 3 Proofs for univariate results

Here we present the proof for Theorem 1. We consider the estimator

$$\widehat{\boldsymbol{d}} \leftarrow \underset{\boldsymbol{d} \in \mathbb{R}^K}{\arg\min} \frac{1}{2n} \|\boldsymbol{y} - RW^\top \boldsymbol{d}\|_2^2 + \lambda \|\boldsymbol{d}_M\|_1, \tag{1}$$

where $\boldsymbol{d}_M$ denotes the sub-vector corresponding to the mother wavelet coefficients. We use this notation to generalize the case of $j_0 = 0$ where $j_0$ denotes the minimum resolution level. One nice feature about (1) is that it is exactly the lasso problem [Tibshirani, 1996] with design matrix $RW^\top$.

*Proof of Theorem 1.* We can divide the proof into three parts, (1) the deterministic part, (2) the stochastic part and (3) the approximation error part. The first 2 parts are standard in the lasso literature, for this reason we will use the results from the book by van de Geer [2016].

**Deterministic Part**

As per Theorem 2.1 of van de Geer [2016] let $\lambda_\varepsilon$ satisfy

$$\lambda_\varepsilon \geq \|WR^\top \boldsymbol{\varepsilon}\|_\infty / n,$$

where $\boldsymbol{\varepsilon}$ is the noise vector. Define for $\lambda > \lambda_\varepsilon$

$$\overline{\lambda} = \lambda + \lambda_\varepsilon, \quad \underline{\lambda} = \lambda - \lambda_\varepsilon,$$

and stretching factor $L = \overline{\lambda}/\underline{\lambda}$. Further more, for an index set $S \subset \{1, \dots, K\}$ and stretching factor $L$ define the *compatibility constant* as

$$\widehat{\vartheta}^2(L, S) = \min \left\{ n^{-1} |S| \|RW^\top \boldsymbol{d}\|_2^2 : \|\boldsymbol{d}_S\|_1 = 1, \|\boldsymbol{d}_{-S}\|_1 \leq L \right\}, \tag{2}$$

where $\boldsymbol{d}_S$ is the vector $\boldsymbol{d}$ with values equal to 0 for indices in $S$. Similarly $\boldsymbol{d}_{-S}$ is the vector $\boldsymbol{d}$ with values equal to 0 for indices in $S^c$. Then we have for any set $S$, and vector $\boldsymbol{d}^*$ we have

$$n^{-1} \|\widehat{\boldsymbol{f}} - \boldsymbol{f}^0\|_2^2 \leq n^{-1} \|\boldsymbol{f}^0 - RW^\top \boldsymbol{d}^*\|_2^2 + \frac{|S|\overline{\lambda}^2}{\widehat{\vartheta}^2(L, S)}. \tag{3}$$

For simplicity we take the $\lambda = 2\lambda_\varepsilon$ giving us $\overline{\lambda} = 3\lambda_\varepsilon$, $\underline{\lambda} = \lambda_\varepsilon$ and $L = 3$. $\qquad\square$

We consider a quick calculation of the compatibility constant $\widehat{\vartheta}(L, S)$. Let $\Lambda_{\min}(R)$ be the minimum eigenvalue of $R$, this will normally be greater than 0 if $K < n$. We then note that:

$$
\begin{aligned}
n^{-1}|S|\|RW^\top \boldsymbol{d}\|_2^2 &\geq \Lambda_{\min}(R) n^{-1} |S| \|\boldsymbol{d}\|_2^2 \\
&= \Lambda_{\min}(R) n^{-1} |S| \left\{ \|\boldsymbol{d}_S\|_2^2 + \|\boldsymbol{d}_{-S}\|_2^2 \right\} \\
&\geq \Lambda_{\min}(R) n^{-1} |S| \left\{ \frac{\|\boldsymbol{d}_S\|_1^2}{|S|} + \frac{\|\boldsymbol{d}_{-S}\|_1^2}{K - |S|} \right\},
\end{aligned}
$$

and minimizing the right hand side under the constraints $\|\boldsymbol{d}_S\|_1 = 1$ and $\|\boldsymbol{d}_{-S}\|_1 \leq L$ we can get that it is bounded below by $\Lambda_{\min}(R) n^{-1}$. This gives us one possible value for the compatibility constant $\widehat{\vartheta}^2(L, S)$, notice that this includes the special case of traditional wavelet regression with $R = I$ and $\Lambda_{\min}(R) = 1$.

Thus we have that

$$n^{-1} \|\widehat{\boldsymbol{f}} - \boldsymbol{f}^0\|_2^2 \leq n^{-1} \|\boldsymbol{f}^0 - RW^\top \boldsymbol{d}^*\|_2^2 + \frac{9n|S|\lambda_\varepsilon^2}{\Lambda_{\min}(R)}. \tag{4}$$

**Stochastic part**

We focus on obtaining a possible values for $\lambda_\varepsilon$. We start with the simple case where $R = I$ and $\boldsymbol{\varepsilon} \sim \mathcal{N}(0, \sigma^2 I)$, i.e. the traditional wavelet approach with regularly spaced data. In this case we need to find a $\lambda_\varepsilon$ such that

$$\lambda_\varepsilon \geq \|W\boldsymbol{\varepsilon}\|_\infty / n. \tag{5}$$

First note that $\boldsymbol{\varepsilon}' = W\boldsymbol{\varepsilon}/\sigma \sim \mathcal{N}(0, I)$ by orthogonality of $W$. Hence we have

$$Pr\left(\|\boldsymbol{\varepsilon}'\|_\infty > \sqrt{t^2 + 2\log n}\right) \leq 2p\exp\left[-\frac{t^2 + 2\log p}{2}\right] = 2\exp(-t^2/2). \tag{6}$$

Thus with probability at-least $1 - 2\exp(-t^2/2)$ we have $\sigma\sqrt{t^2 + 2\log n} \geq \|W\boldsymbol{\varepsilon}\|_\infty$. Thus in this case we can take $\lambda_\varepsilon = n^{-1}\sigma\sqrt{t^2 + 2\log n}$. In the general case we would have the mean zero, sub-Gaussian $K$-vector $WR^\top\boldsymbol{\varepsilon}$. By a slightly more involved argument we can show that we can take $\lambda_\varepsilon = n^{-1}c_1\sqrt{t^2 + 2\log K}$ where $c_1$ depends on the distribution of $\varepsilon$ (i.e., the parameters of the sub-gaussian distribution) and matrix $R$.

Thus we have shown so far that with probability at-least $1 - 2\exp(-t^2/2)$ we have

$$n^{-1}\|\widehat{\boldsymbol{f}} - \boldsymbol{f}^0\|_2^2 \leq n^{-1}\|\boldsymbol{f}^0 - RW^\top\boldsymbol{d}^*\|_2^2 + \frac{9c_1^2}{\Lambda_{\min}(R)}\frac{|S|(t^2 + 2\log K)}{n}, \tag{7}$$

or without worrying about optimal constants we get the rate

$$n^{-1}\|\widehat{\boldsymbol{f}} - \boldsymbol{f}^0\|_2^2 \leq n^{-1}\|\boldsymbol{f}^0 - RW^\top\boldsymbol{d}^*\|_2^2 + C\frac{|S|\log K}{n}. \tag{8}$$

To obtain our result we just need the final step: approximation error.

**Approximation error part**

Now we will bound the term $n^{-1}\|\boldsymbol{f}^0 - RW^\top\boldsymbol{d}^*\|_2^2$. We will define specific types of vectors $\boldsymbol{d}^*$ which leads to specific sparse indes sets $S$. We begin with the decomposition:

$$n^{-1}\|\boldsymbol{f}^0 - RW^\top\boldsymbol{d}^*\|_2^2 \leq 2n^{-1}\|\boldsymbol{f}^0 - R\widetilde{\boldsymbol{f}}^0\|_2^2 + 2n^{-1}\|R\widetilde{\boldsymbol{f}}^0 - RW^\top\boldsymbol{d}^*\|_2^2, \tag{9}$$

where $\widetilde{f}^0$ is the function obtained by interpolating $f^0$ from the data $(i/K, f^0(i/K))$ for $i = 1, \ldots, K$ and $\widetilde{\boldsymbol{f}}^0 = [\widetilde{f}^0(1/K), \ldots, \widetilde{f}^0(K/K)]^\top$.

For the second term, define $\Lambda_{\max}(R)$ as the maximum eigenvalue of $R^\top R$ then

$$n^{-1}\|R\widetilde{\boldsymbol{f}}^0 - RW^\top\boldsymbol{d}^*\|_2^2 \leq \Lambda_{\max}(R)n^{-1}\|\widetilde{\boldsymbol{f}}^0 - W^\top\boldsymbol{d}^*\|_2^2 \leq \Lambda_{\max}(R)\|\widetilde{\boldsymbol{f}}^0 - W^\top\boldsymbol{d}^*\|_\infty^2.$$

For the last part we now define $\boldsymbol{d}^*$, the vector of wavelet coefficients such that it defines a function $f^*$ as a linear combination of wavelet basis functions. To be precise we have that

$$f^*(x) = \sum_{k=0}^{2^{j_0}-1} \phi_{j_0 k}(x)\alpha_{j_0 k}^0 + \sum_{j=j_0}^{J^*-1}\sum_{k=0}^{2^j-1} \psi_{jk}(x)\beta_{jk}^0, \tag{10}$$

for some integer $J^*$, and where $\alpha_{j0k}^0$ and $\beta_{jk}^0$ are the wavelet coefficients of the true function $f^0$. Now we obtain:

$$\max_x |f^*(x) - f^0(x)| = \max_x \left|\sum_{j=J^*}^{\infty}\sum_{k=0}^{2^j-1} \psi_{jk}(x)\beta_{jk}^0\right|$$

$$\leq \max_x \max_{j \geq J^*, k} |\psi_{jk}(x)| \sum_{j=J^*}^{\infty}\sum_{k=0}^{2^j-1} |\beta_{jk}^0|$$

$$= \max_x \max_{j \geq J^*, k} |\psi_{jk}(x)| \sum_{j=J^*}^{\infty} \|\boldsymbol{\beta}_j^0\|_1,$$

where $\boldsymbol{\beta}_j \in \mathbb{R}^{2^j}$ is the mother wavelet coefficient vector at level $j$. Now assuming that $f^0 \in B_{q_1, q_2}^s$

$$\sum_{j=J^*}^{\infty} \|\boldsymbol{\beta}_j^0\|_1 = \sum_{j=J^*}^{\infty} \frac{2^{js'}}{2^{js'}}\|\boldsymbol{\beta}_j^0\|_1, \qquad (s' = s - 1/2)$$

$$\leq \left[\sum_{j=J^*}^{\infty}\left(2^{js'}\|\boldsymbol{\beta}_j^0\|_1\right)^{q_2}\right]^{1/q_2}\left[\sum_{j=J^*}^{\infty} 2^{-js'q_2'}\right]^{1/q_2'},$$

where $q_2'$ is such that $1/q_2 + 1/q_2' = 1$. Using the inequality $\|\boldsymbol{\beta}_j^0\|_1 \leq 2^{j(1-1/q_1)}\|\boldsymbol{\beta}_j^0\|_{q_1}$ we get

$$\sum_{j=J^*}^{\infty} \|\boldsymbol{\beta}_j^0\|_1 \leq \left[ \sum_{j=J^*}^{\infty} \left( 2^{j(s+1/2-1/q_1)}\|\boldsymbol{\beta}_j^0\|_{q_1} \right)^{q_2} \right]^{1/q_2} \left[ \sum_{j=J^*}^{\infty} 2^{-js'q_2'} \right]^{1/q_2'}$$

$$= \left[ \sum_{j=J^*}^{\infty} \left( 2^{j(s+1/2-1/q_1)}\|\boldsymbol{\beta}_j^0\|_{q_1} \right)^{q_2} \right]^{1/q_2} \times C_2 2^{-J^*s},$$

where the second term can be obtained by looking at $S_\infty - S_{J^*-1}$ where $S_n = \sum_{j=0}^{n} 2^{-js'q_2'}$. The first term is bounded because $f^0 \in B_{q_1,q_2}^s$.

**Putting the pieces together**

Thus we have shown so far, by taking $\boldsymbol{d}^*$ as defined above and $S$ being the active set of $\boldsymbol{d}^*$ (i.e. $|S| = 2^{J^*}$), that the rate is of the form (upto constants)

$$n^{-1}\|\widehat{\boldsymbol{f}} - \boldsymbol{f}^0\|_2^2 \leq 2n^{-1}\|\boldsymbol{f}^0 - R\widetilde{\boldsymbol{f}}^0\|_2^2 + C_2 2^{-(2s)J^*} + C_3 2^{J^*} \frac{\log K}{n}.$$

Treating the above as a function of $J^*$ and minimizing we obtain the approximate truncation order $|S| = \mathcal{O}(n^{1/(2s+1)})$ which minimizes the right hand side. Finally, putting all the different pieces together we obtain the bound:

$$n^{-1}\|\widehat{\boldsymbol{f}} - \boldsymbol{f}^0\|_2^2 \leq C_4 \left( \frac{\log K}{n} \right)^{\frac{2s}{2s+1}} + 2n^{-1}\|\boldsymbol{f}^0 - R\widetilde{\boldsymbol{f}}^0\|_2^2.$$

## 4 Proofs for additive `waveMesh`

### 4.1 Initial results

We will present results in greater generality here. In the interest of brevity and clarity of exposition we avoided some technical details such as identifiablity and the intercept term in the model. We go into these details here.

Let $f^*$ be a sparse additive approximation to $f^0$,

$$f^*(\boldsymbol{x}_i) = c^0 + \sum_{j=1}^{p} f_j^*(x_{ij}) = c^0 + \sum_{j \in S} f_j^*(x_{ij}),$$

where $S = \{j : f_j^* \neq 0\}$, which we call the active set, is a subset of $\{1, \ldots, p\}$ of size $|S|$ and, $c^0 = E(\bar{\boldsymbol{y}})$ where $\bar{\boldsymbol{y}}$ is the sample mean. To ensure identifiability, we assume $\sum_{i=1}^{n} f_j^*(x_{ij}) = 0$ $(j = 1, \ldots, p)$.

We consider a large class of estimators of the type:

$$\widehat{f}_1, \ldots, \widehat{f}_p = \underset{(f_j)_{j=1}^p \in \mathcal{F}}{\arg\min} \frac{1}{2n} \sum_{i=1}^{n} \left\{ y_i - \bar{\boldsymbol{y}} - \sum_{j=1}^{p} f_j(x_{ij}) \right\}^2 + \lambda_n \sum_{j=1}^{p} I(f_j), \qquad (11)$$

where $I(\cdot)$ is a penalty of the form $I(f_j) = \|f_j\|_n + \lambda_n \Upsilon(f_j)$, for a semi-norm $\Upsilon(\cdot)$ and, empirical norm $\|\cdot\|_n$ defined for component $f_j$ as $\|f_j\|_n^2 = n^{-1}\sum_{i=1}^{n}[f_j(x_{ij})]^2$. In our case $\Upsilon(\cdot)$ is the Besov norm of the $B_{1,1}^s$ space.

Throughout this proof, instead of the smoothness level $s$, we will use $\alpha = 1/s$. Before we begin the main proof, we define the notion of metric entropy which will be used throughout the proof. For a set $\mathcal{F}$ equipped with some metric $d(\cdot, \cdot)$, the subset $\{f_1, \ldots, f_N\} \subset \mathcal{F}$ is a $\delta$-cover if for any $f \in \mathcal{F}$ $\min_{1 \leq i \leq N} d(f, f_i) \leq \delta$. The log-cardinality of the smallest $\delta$-cover is the $\delta$-entropy of $\mathcal{F}$ with respect to metric $d(\cdot, \cdot)$. We denote by $H(\delta, \mathcal{F}, Q)$, the $\delta$-entropy of a function class $\mathcal{F}$ with respect to the $\|\cdot\|_Q$ metric for a measure $Q$, where $\|f\|_Q^2 = \int \{f(x)\}^2 dQ(x)$. For a fixed sample of

covariate $j$, $x_{1j}, \ldots, x_{nj}$, we denote by $Q_{nj}$ the empirical measure $Q_{nj} = n^{-1} \sum_{i=1}^{n} \delta_{x_{ij}}$ and use the short-hand notation $\| \cdot \|_n = \| \cdot \|_{Q_{nj}}$.

The main ingredient we require for proving results for sparse additive models is the entropy condition, specifically we require

$$H(\delta, \{f_j \in \mathcal{F} : \Upsilon(f_j) \leq 1\}, Q_{nj}) \leq A_0 \delta^{-\alpha},$$

for $\alpha \in (0, 2)$, and so forth.

**Note:** In the case of our Besov norm, the above entropy condition holds for $\alpha = 1/s$, i.e.,

$$H(\delta, \{f_j \in \mathcal{F} : \Upsilon(f_j) \leq 1\}, Q_{nj}) \leq A_0 \delta^{-1/s}.$$

**Lemma 1** (Basic inequality). *For any function $f^* = \sum_{j=1}^{p} f_j^*$, where $f_j^* \in \mathcal{F}$ and, the solution $\widehat{f}$ of (11), we have the following basic inequality*

$$\frac{1}{2}\|\widehat{f} - f^0\|_n^2 + \lambda I_p(\widehat{f}) \leq |\langle \varepsilon, \widehat{f} - f^* \rangle_n| + \lambda I_p(f^*) + |\bar{\varepsilon}| \sum_{j=1}^{p} \|\widehat{f_j} - f_j^*\|_n + \frac{1}{2}\|f^* - f^0\|_n^2,$$

*where $\langle \varepsilon, f \rangle_n = \frac{1}{n} \sum_{i=1}^{n} \varepsilon_i f(x_i)$, $\bar{\varepsilon} = \frac{1}{n} \sum_{i=1}^{n} \varepsilon_i$ and $I_p(f) = \sum_{j=1}^{p} I(f_j) = \sum_{j=1}^{p} \|f_j\|_n + \lambda \Upsilon(f_j)$ for an additive function $f$.*

*Proof.* We have

$$\frac{1}{2n} \sum_{i=1}^{n} \left\{ y_i - \bar{y} - \widehat{f}(x_i) \right\}^2 + \lambda I_p(\widehat{f}) \leq \frac{1}{2n} \sum_{i=1}^{n} \left\{ y_i - \bar{y} - f^*(x_i) \right\}^2 + \lambda I_p(f_j^*),$$

$$\Leftrightarrow \frac{1}{2n} \sum_{i=1}^{n} \left\{ \varepsilon_i + c^0 - \bar{y} - (\widehat{f} - f^0)(x_i) \right\}^2 + \lambda I_p(\widehat{f}) \leq \frac{1}{2n} \sum_{i=1}^{n} \left\{ \varepsilon_i + c^0 - \bar{y} - (f^* - f^0)(x_i) \right\}^2 + \lambda I_p(f_j^*)$$

$$\Rightarrow \frac{1}{2n} \sum_{i=1}^{n} \left( \varepsilon_i + c^0 - \bar{y} \right)^2 + (\widehat{f} - f^0)^2(x_i) - 2(\varepsilon_i + c^0 - \bar{y})(\widehat{f} - f^0)(x_i) + \lambda I_p(\widehat{f})$$

$$\leq \frac{1}{2n} \sum_{i=1}^{n} \left( \varepsilon_i + c^0 - \bar{y} \right)^2 + (f^* - f^0)^2(x_i) - 2(\varepsilon_i + c^0 - \bar{y})(f^* - f^0)(x_i) + \lambda I_p(f^*)$$

$$\Rightarrow \frac{1}{2}\|\widehat{f} - f^0\|_n^2 - \langle \varepsilon + c^0 - \bar{y}, \widehat{f} - f^0 \rangle_n + \lambda I_p(\widehat{f})$$

$$\leq \frac{1}{2}\|f^* - f^0\|_n^2 - \langle \varepsilon + c^0 - \bar{y}, f^* - \widehat{f} + \widehat{f} - f^0 \rangle_n + \lambda I_p(f^*)$$

$$\Rightarrow \frac{1}{2}\|\widehat{f} - f^0\|_n^2 - \langle \varepsilon + c^0 - \bar{y}, \widehat{f} - f^0 \rangle_n + \lambda I_p(\widehat{f})$$

$$\leq \frac{1}{2}\|f^* - f^0\|_n^2 - \langle \varepsilon + c^0 - \bar{y}, f^* - \widehat{f} \rangle_n - \langle \varepsilon + c^0 - \bar{y}, \widehat{f} - f^0 \rangle_n + \lambda I_p(f^*),$$

which implies

$$\frac{1}{2}\|\widehat{f} - f^0\|_n^2 + \lambda I_p(\widehat{f}) \leq \frac{1}{2}\|f^* - f^0\|_n^2 - \langle \varepsilon + c^0 - \bar{y}, f^* - \widehat{f} \rangle_n + \lambda I_p(f^*)$$

$$\Rightarrow \frac{1}{2}\|\widehat{f} - f^0\|_n^2 + \lambda I_p(\widehat{f}) \leq |\langle \varepsilon, \widehat{f} - f^* \rangle_n| + \sum_{j=1}^{p} \langle c^0 - \bar{y}, \widehat{f_j} - f_j^* \rangle_n + \lambda I_p(f^*) + \frac{1}{2}\|f^* - f^0\|_n^2$$

$$\Rightarrow \frac{1}{2}\|\widehat{f} - f^0\|_n^2 + \lambda I_p(\widehat{f}) \leq |\langle \varepsilon, \widehat{f} - f^* \rangle_n| + |c^0 - \bar{y}| \sum_{j=1}^{p} \|\widehat{f_j} - f_j^*\|_n + \lambda I_p(f^*) + \frac{1}{2}\|f^* - f^0\|_n^2.$$

Now for the second term note that:

$$|c^0 - \bar{y}| = \left| \frac{1}{n} \sum_{i=1}^{n} (c^0 - y_i) \right| = \left| \frac{1}{n} \sum_{i=1}^{n} \left\{ c^0 - c^0 - \sum_{j=1}^{p} f_j^0(x_{i,j}) - \varepsilon_i \right\} \right| = |\bar{\varepsilon}|.$$

Which leads us to

$$\frac{1}{2} \|\widehat{f} - f^0\|_n^2 + \lambda I_p(\widehat{f}) \leq |\langle \varepsilon, \widehat{f} - f^* \rangle_n| + \lambda I_p(f^*) + |\bar{\varepsilon}| \sum_{j=1}^{p} \|\widehat{f}_j - f_j^*\|_n + \frac{1}{2} \|f^* - f^0\|_n^2.$$

$\square$

**Lemma 2** (Bounding the term $|\bar{\varepsilon}|$)**.** *For $\varepsilon = (\varepsilon_1, \ldots \varepsilon_n)^T$ such that $E(\varepsilon_i) = 0$ and*

$$L^2 \left\{ E\left( e^{\varepsilon_i^2/L^2} \right) - 1 \right\} \leq \sigma_0^2 \,,$$

*for all $\kappa > 0$ and*

$$\rho = \kappa \max \left\{ n^{-\frac{1}{2+\alpha}}, \left( \frac{\log p}{n} \right)^{1/2} \right\},$$

*we have that with probability at least $1 - 2\exp\left( -n\rho^2/c_1 \right)$,*

$$|\bar{\varepsilon}| \leq \rho,$$

*for a constant $c_1$ that depends on $L$ and $\sigma_0$.*

*Proof.* By Lemma 8·2 of van de Geer [2000] (with $\gamma_n = 1_n/n$) we have for all $t > 0$

$$\mathrm{pr}\left( \left| \frac{1}{n} \sum_{i=1}^{n} \varepsilon_i \right| \geq t \right) \leq 2\exp\left\{ -\frac{nt^2}{8(L^2 + \sigma_0^2)} \right\}.$$

The result follows by setting $t = \rho$. $\square$

**Lemma 3** (Bounding the term $|\langle \varepsilon, \widehat{f} - f^* \rangle_n|$)**.** *For $\lambda \geq 4\rho$ where*

$$\rho = \kappa \max \left\{ n^{-\frac{1}{2+\alpha}}, \left( \frac{\log p}{n} \right)^{1/2} \right\},$$

*for some constant $\kappa$, if*

$$H(\delta, \{f \in \mathcal{F} : \Upsilon(f) \leq 1\}, Q_n) \leq A_0 \delta^{-\alpha},$$

*we then have with probability at least $1 - c_2 \exp\left( -c_3 n\rho^2 \right)$*

$$|\langle \varepsilon, \widehat{f}_j - f_j^* \rangle_n| \leq \rho \|\widehat{f}_j - f_j^*\|_n + \rho\lambda \Upsilon(\widehat{f}_j - f_j^*),$$

*for all $j = 1, \ldots, p$ and positive constants $c_2$ and $c_3$.*

*Proof.* Firstly, for $\mathcal{F}_0 = \{f \in \mathcal{F} : \Upsilon(f) \leq 1\}$ we have by assumption a $\delta$ cover $f_1, \ldots, f_N$ such that for all $f \in \mathcal{F}_0$ we have $\min_{j \in \{1,\ldots,N\}} \|f_j - f\|_n \leq \delta$. Now we are interested in the set $\mathcal{F}_{0,\lambda} = \{f \in \mathcal{F} : \lambda \Upsilon(f) \leq 1\}$. Firstly, for a function $f \in \mathcal{F}_{0,\lambda}$,

$$\min_{j \in \{1,\ldots,N\}} \|f - f_j/\lambda\|_n = \min_{j \in \{1,\ldots,N\}} \frac{1}{\lambda} \|\lambda f - f_j\|_n \leq \frac{\delta}{\lambda},$$

because $\Upsilon(\lambda f) = \lambda \Upsilon(f) \leq 1 \Rightarrow \lambda f \in \mathcal{F}_0$. This means that the set $\{f_1/\lambda, \ldots, f_N/\lambda\}$ is a $\delta/\lambda$ cover of the set $\mathcal{F}_{0,\lambda}$.

This implies that $H(\delta, \mathcal{F}_0, Q_n) \leq A_0 \delta^{-\alpha} \Rightarrow H(\delta/\lambda, \mathcal{F}_{0,\lambda}, Q_n) \leq A_0 \delta^{-\alpha}$ or equivalently $H(\delta, \mathcal{F}_{0,\lambda}, Q_n) \leq A_0 (\delta\lambda)^{-\alpha}$. Finally, since $\{f \in \mathcal{F} : I(f) \leq 1\} \subset \{f \in \mathcal{F} : \Upsilon(f) \leq \lambda^{-1}\}$ we have

$$H(\delta, \{f \in \mathcal{F} : I(f) \leq 1\}, Q_n) \leq A_0 (\delta\lambda)^{-\alpha}.$$

The same entropy bound holds for the class

$$\widetilde{\mathcal{F}} = \left\{ \frac{f_j - f_j^*}{\|f_j - f_j^*\|_n + \lambda \Upsilon(f_j - f_j^*)} : f_j \in \mathcal{F} \right\}, \tag{12}$$

and we can now apply Corollary 8.3 of van de Geer [2000] by noting that

$$\int_0^1 H^{1/2}(u, \widetilde{\mathcal{F}}, Q_n)\, du \leq \widetilde{A}_0 \lambda^{-\alpha/2},$$

for some constant $\widetilde{A}_0 = \widetilde{A}_0(A_0)$. For some $c_2 = c_2(L, \sigma_0)$ and all $\delta \geq 2c_2\widetilde{A}_0\lambda^{-\alpha/2}n^{-1/2}$ we have

$$\text{pr}\left( \sup_{f_j \in \mathcal{F}} \frac{|\langle \varepsilon, f_j - f_j^* \rangle_n|}{\|f_j - f_j^*\|_n + \lambda\Upsilon(f_j - f_j^*)} \geq \delta \right) \leq c_2 \exp\left( -\frac{n\delta^2}{4c_2^2} \right). \tag{13}$$

Since $\lambda \geq \rho$ we note that $2c_2\widetilde{A}_0\lambda^{-\alpha/2}n^{-1/2} \leq 2c_2\widetilde{A}_0\rho^{-\alpha/2}n^{-1/2}$ and that

$$2c_2\widetilde{A}_0\rho^{-\alpha/2}n^{-1/2} \leq \rho \Leftrightarrow \rho \geq \left( 2c_2\widetilde{A}_0 \right)^{\frac{2}{2+\alpha}} n^{-\frac{1}{2+\alpha}}.$$

Which holds by definition since $\rho = \kappa \max\left\{ (\log p/n)^{1/2}, n^{-1/(2+\alpha)} \right\} \geq \kappa n^{-1/(2+\alpha)}$ and $\kappa$ is sufficiently large (any $\kappa \geq \left( 2c_2\widetilde{A}_0 \right)^{2/(2+\alpha)}$ would suffice). Therefore, we can take $\delta = \rho$ in (13) along with a union bound to obtain

$$\text{pr}\left( \max_{j=1,\ldots,p} \sup_{f_j \in \mathcal{F}} \frac{|\langle \varepsilon, f_j - f_j^* \rangle_n|}{\|f_j - f_j^*\|_n + \lambda\Upsilon(f_j - f_j^*)} \geq \rho \right) \leq pc_2 \exp\left( -\frac{n\rho^2}{4c_2^2} \right)$$

$$= c_2 \exp\left\{ -n\rho^2 \left( \frac{1}{4c_2^2} - \frac{\log p}{n\rho^2} \right) \right\}$$

$$\leq c_2 \exp\left( -n\rho^2 c_3 \right),$$

for some positive constant $c_3 = c_3(c_2, \widetilde{A}_0)$.

Finally, we show that $c_3 > 0$. This follows from the fact that $1/(4c_2^2) - \log p/(n\rho^2) > 0 \Leftrightarrow n\rho^2 > 4c_2^2 \log p$. This holds since $n\rho^2 \geq \kappa^2 \log p$ for $\kappa$ sufficiently large. Thus, we have with probability at least $1 - c_2 \exp\left( c_3 n\rho^2 \right)$ for all $j = 1, \ldots, p$

$$|\langle \varepsilon, \widehat{f}_j - f_j^* \rangle_n| \equiv |\langle \varepsilon, \widehat{\Delta}_j \rangle_n| \leq \rho \|\widehat{\Delta}_j\|_n + \rho\lambda\Upsilon(\widehat{\Delta}_j).$$

$\square$

## 4.2 Using the active set

So far we have shown that, for $\lambda \geq 4\rho$, with probability at least $1 - 2\exp\left( -n\rho/c_1 \right) - c_2 \exp\left( -c_3 n\rho^2 \right)$, the following inequality holds

$$\|\widehat{f} - f^0\|_n^2 + 2\lambda \sum_{j=1}^p I(\widehat{f}_j) \leq 2|\langle \varepsilon, \widehat{f} - f^* \rangle_n| + 2|\bar{\varepsilon}| \sum_{j=1}^p \|\widehat{\Delta}_j\|_n + 2\lambda \sum_{j=1}^p I(f_j^*) + \|f^* - f^0\|_n^2$$

$$\leq \left\{ \sum_{j=1}^p 2\rho\|\widehat{\Delta}_j\|_n + 2\rho\lambda\Upsilon(\widehat{\Delta}_j) \right\} + \left( 2\rho \sum_{j=1}^p \|\widehat{\Delta}_j\|_n \right)$$

$$+ \left\{ 2\lambda \sum_{j=1}^p I(f_j^*) \right\} + \|f^* - f^0\|_n^2$$

$$\Rightarrow \|\widehat{f} - f^0\|_n^2 + 2\lambda \sum_{j=1}^p I(\widehat{f}_j) \leq \sum_{j=1}^p \left\{ \lambda\|\widehat{\Delta}_j\|_n + \frac{\lambda^2}{2}\Upsilon(\widehat{\Delta}_j) + 2\lambda\|f_j^*\|_n + 2\lambda^2\Upsilon(f_j^*) \right\} + \|f^* - f^0\|_n^2.$$

For notational convenience we will exclude the $\|f^* - f^0\|_n^2$ term in the following manipulations. If $S$ is the active set then we have on the right hand side,

$$\text{RHS} = \lambda \sum_{j \in S} \left\{ \|\widehat{\Delta}_j\|_n + \frac{\lambda}{2}\Upsilon(\widehat{\Delta}_j) + 2\|f_j^*\|_n + 2\lambda\Upsilon(f_j^*) \right\} + \lambda \sum_{j \in S^c} \left\{ \|\widehat{f}_j\|_n + \frac{\lambda}{2}\Upsilon(\widehat{f}_j) \right\}$$

$$\leq \lambda \sum_{j \in S} \left\{ \|\widehat{\Delta}_j\|_n + \frac{\lambda}{2}\Upsilon(\widehat{\Delta}_j) + 2\|\widehat{\Delta}_j\|_n + 2\|\widehat{f}_j\|_n + 2\lambda\Upsilon(f_j^*) \right\} + \lambda \sum_{j \in S^c} \left\{ \|\widehat{f}_j\|_n + \frac{\lambda}{2}\Upsilon(\widehat{f}_j) \right\}$$

$$= 3\sum_{j \in S} \lambda\|\widehat{\Delta}_j\|_n + 2\sum_{j \in S} \lambda^2\Upsilon(f_j^*) + \sum_{j \in S^c} \lambda\|\widehat{f}_j\| + \frac{1}{2}\sum_{j \in S^c} \lambda^2\Upsilon(\widehat{f}_j) + 2\sum_{j \in S} \lambda\|\widehat{f}_j\|_n + \frac{1}{2}\sum_{j \in S} \lambda^2\Upsilon(\widehat{\Delta}_j),$$

where the inequality holds by the decomposition $\|f_j^*\|_n = \|f_j^* - \widehat{f}_j + \widehat{f}_j\|_n \leq \|\widehat{\Delta}_j\|_n + \|\widehat{f}_j\|_n$.

On the left hand side we have

$$\text{LHS} = \|\widehat{f} - f^0\|_n^2 + 2\lambda \sum_{j \in S} \left\{ \|\widehat{f}_j\|_n + \lambda\Upsilon(\widehat{f}_j) \right\} + 2\lambda \sum_{j \in S^c} \left\{ \|\widehat{f}_j\|_n + \lambda\Upsilon(\widehat{f}_j) \right\}$$

$$\geq \|\widehat{f} - f^0\|_n^2 + 2\lambda \sum_{j \in S} \left\{ \|\widehat{f}_j\|_n + \lambda\Upsilon(\widehat{\Delta}_j) - \lambda\Upsilon(f_j^*) \right\} + 2\lambda \sum_{j \in S^c} \left\{ \|\widehat{f}_j\|_n + \lambda\Upsilon(\widehat{f}_j) \right\},$$

where the inequality follows from the triangle inequality $\Upsilon(\widehat{f}_j) + \Upsilon(f_j^*) \geq \Upsilon(\widehat{\Delta}_j)$ since $\Upsilon(\cdot)$ is a semi-norm. By re-arranging the terms we obtain the inequality

$$\|\widehat{f} - f^0\|_n^2 + \lambda \sum_{j \in S^c} \left\{ \|\widehat{f}_j\|_n + \frac{3\lambda}{2}\Upsilon(\widehat{f}_j) \right\} + \frac{3\lambda^2}{2} \sum_{j \in S} \Upsilon(\widehat{\Delta}_j) \leq 3\lambda \sum_{j \in S} \|\widehat{\Delta}_j\|_n + 4\lambda^2 \sum_{j \in S} \Upsilon(f_j^*) + \|f^* - f^0\|_n^2$$

which implies that

$$\|\widehat{f} - f^0\|_n^2 + \lambda \sum_{j \in S^c} \|\widehat{\Delta}_j\|_n + \frac{3\lambda^2}{2} \sum_{j=1}^p \Upsilon(\widehat{\Delta}_j) \leq 3\lambda \sum_{j \in S} \|\widehat{\Delta}_j\|_n + 4\lambda^2 \sum_{j \in S} \Upsilon(f_j^*) + \|f^* - f^0\|_n^2.$$

This implies the slow rates for convergence for $\lambda \geq 4\rho$ and $|S|$

$$\frac{1}{2}\|\widehat{f} - f^0\|_n^2 + \leq |S|\lambda \left\{ 3\sum_{j \in S} \|\widehat{\Delta}_j\|_n/|S| + 2\lambda \sum_{j \in S} \Upsilon(f_j^*)/|S| \right\} + \frac{1}{2}\|f^* - f^0\|_n^2.$$

This completes the proof of the first part of the theorem. Recall that $\lambda$ is of the order:

$$\kappa \max \left\{ n^{-\frac{1}{2+\alpha}}, \left(\frac{\log p}{n}\right)^{1/2} \right\},$$

and for the Besov space $B_{1,1}^s$ we have $\alpha = 1/s$.

## 4.3   Using the compatibility condition

Recall the compatibility condition for $f = \sum_{j=1}^p f_j$, whenever

$$\sum_{j \in S^c} \|f_j\|_n + \frac{3\lambda}{2} \sum_{j=1}^p \Upsilon(f_j) \leq 3\sum_{j \in S} \|f_j\|_n, \tag{14}$$

then we have

$$\sum_{j \in S} \|f_j\|_n \leq |S|^{1/2}\|f\|_n/\vartheta(S).$$

Once we assume the compatibility condition we can prove the rest of the theorem by considering the following two cases.

**Case 1:** $\lambda \sum_{j \in S} \|\widehat{\Delta}_j\|_n \geq 4\lambda^2 \sum_{j \in S} \Upsilon(f_j^*)$ in which case we have

$$\|\widehat{f} - f^0\|_n^2 + \lambda \sum_{j \in S^c} \|\widehat{\Delta}_j\|_n + \frac{3\lambda^2}{2} \sum_{j=1}^p \Upsilon(\widehat{\Delta}_j) \leq 4\lambda \sum_{j \in S} \|\widehat{\Delta}_j\|_n + \|f^* - f^0\|_n^2 \,,$$

hence for the function $\widehat{f} - f^* = \sum_{j=1}^p \widehat{\Delta}_j$ (14) holds and hence by the compatibility condition we have

$$\|\widehat{f} - f^0\|_n^2 + \lambda \sum_{j \in S^c} \|\widehat{\Delta}_j\|_n + \frac{3\lambda^2}{2} \sum_{j=1}^p \Upsilon(\widehat{\Delta}_j) \leq \frac{4\lambda |S|^{1/2}}{\vartheta(S)} \|\widehat{f} - f^*\|_n + \|f^* - f^0\|_n^2$$

$$\leq \frac{4\lambda |S|^{1/2}}{\vartheta(S)} \|\widehat{f} - f^0\|_n + \frac{4\lambda |S|^{1/2}}{\vartheta(S)} \|f^* - f^0\|_n + \|f^* - f^0\|_n^2$$

$$\leq 2 \left\{ \frac{2\lambda (2s)^{1/2}}{\vartheta(S)} \right\} \left( \frac{\|\widehat{f} - f^0\|_n}{2^{1/2}} \right) + 2 \left\{ \frac{2\lambda |S|^{1/2}}{\vartheta(S)} \right\} \left( \|f^* - f^0\|_n \right) + \|f^* - f^0\|_n^2$$

$$\leq \frac{4\lambda^2 (2|S|)}{\vartheta^2(S)} + \frac{\|\widehat{f} - f^0\|_n^2}{2} + \frac{4\lambda^2 |S|}{\vartheta^2(S)} + \|f^* - f^0\|_n^2 + \|f^* - f^0\|_n^2$$

$$\leq \frac{12\lambda^2 |S|}{\vartheta^2(S)} + \frac{\|\widehat{f} - f^0\|_n^2}{2} + 2\|f^* - f^0\|_n^2,$$

where we use the inequality $2ab \leq a^2 + b^2$ and this implies that

$$\frac{1}{2}\|\widehat{f} - f^0\|_n^2 + \lambda \sum_{j \in S^c} \|\widehat{\Delta}_j\|_n + \frac{3\lambda^2}{2} \sum_{j=1}^p \Upsilon(\widehat{\Delta}_j) \leq \frac{12s\lambda^2}{\vartheta^2(S)} + 2\|f^* - f^0\|_n^2.$$

**Case 2:** $\lambda \sum_{j \in S} \|\widehat{\Delta}_j\|_n \leq 4\lambda^2 \sum_{j \in S} \Upsilon(f_j^*)$ in which case we have

$$\|\widehat{f} - f^0\|_n^2 + \lambda \sum_{j \in S^c} \|\widehat{\Delta}_j\|_n + \frac{3\lambda^2}{2} \sum_{j=1}^p \Upsilon(\widehat{\Delta}_j) \leq 16\lambda^2 \sum_{j \in S} \Upsilon(f_j^*) + \|f^* - f^0\|_n^2$$

$$\leq 16|S|\lambda^2 \sum_{j \in S} \Upsilon(f_j^*)/|S| + \|f^* - f^0\|_n^2,$$

which implies

$$\frac{1}{2}\|\widehat{f} - f^0\|_n^2 + \lambda \sum_{j \in S^c} \|\widehat{\Delta}_j\|_n + \frac{3\lambda^2}{2} \sum_{j \in S} \Upsilon(\widehat{\Delta}_j) \leq 16|S|\lambda^2 \sum_{j \in S} \Upsilon(f_j^*)/|S| + 2\|f^* - f^0\|_n^2.$$