[Reviews · NeurIPS 2018]

Reviewer 1



Summary: The paper considers the problem of non-parametric regression using Wavelet basis functions. The main contribution of the paper is a clever interpolation scheme that allows the authors to handle irregularly spaced data. With the interpolation scheme the estimation problem reduces to a Lasso type problem and the authors are able to use existing technical machinery developed for the standard Lasso problem to obtain minimax convergence rates. Comments: The paper is well written and easy to follow. The interpolation method is novel and appears to be useful in practice. However, the Theoretical analysis is essentially standard. The proposal of using additive models for high-dimensional estimation is also fairly standard. I have essentially some minor comments to improve the presentation of the paper. 1) The linear interpolation scheme (lines 117 to 118) can be described a bit more intuitively. 2) The compatibility condition for wavelet regression can also be explained a bit more intuitively.

Reviewer 2



This paper proposes regression methods using wavelets, which does not require irregularly spaced data or the number of observation to be a power of 2. The key idea is to interpolate the raw data to fitted values on the regular grid and run regression algorithm with l1-penalty, i.e., proximal gradient descent. It is natural to generalize additive models for a potentially large number of covariates. The authors analyze (minimax) convergence rates of the proposed methods over Besov spaces. In experiments, they benchmark their methods under some synthetic functions and report that they perform better than competitors. The necessary assumptions of traditional wavelets regressions (equi-spaced and a power of 2 data) restricts to apply them into practical applications. Therefore, it is nice work as authors overcome this issue. The analysis for convergence rates justifies to solve l1-penelty optimization. However, the interpolation idea is very natural to deal with irregular data points. More detail comparisons with other interpolating works would be required. In addition, the experiments are only benchmarked with a power of 2 observations. This is not enough to demonstrate that the proposed methods are practically applicable. Authors should report the cases with general number of observations. In addition, although the experiments works well in some artificial functions (e.g., polynomial, sine and etc), a lack of benchmarks under real-world functions makes the paper weaken. It is worth to report additional experiments under real-world regression problems. It would be better to understand if authors give more intuitions of interpolation schemes, e.g., construction of R in section 2.2, and how this can be computed by providing quantitative complexity in section 2.3. In addition, some readers may wonder how the linear interpolation matrix R is defined in Theorem 1. The minor typos: In line 76: \phi_{jk}(x) = 2^{j/2} “”\phi””(2^j x - k) In line 143: O(K log k) -> O(K log K) In line 177: ca -> can

Reviewer 3



The key contribution of the paper is a method to adapt wavelet shrinkage to irregularly spaced covariates x. The idea is to solve for wavelet coefficients d on a regular K=2^J size grid while minimizing the wavelet shrinkage loss where the fitted values at the given covariates are interpolated from the fitted values W'd on the regular grid by a linear transformation R that depends on x. A related strategy in the literature is to minimize \| Sy - W'd\|_2^2 + \lambda \|d\|_1 where S interpolates y from R^n to the regular grid and W' is the wavelet transform. But in this paper, they minimize \| y - RW'd \|_2^2 + \lambda \|d\|_1 where R interpolates the fitted values on the regular grid to R^n. The former method takes only O(n) time after sorting (and writing it as \|WSy-d\|_2^2 etc.). The latter is slower because we have to solve a lasso type of problem. Antoniadis and Fan (2001) also extend wavelets to irregular data in a similar fashion. However, their method is restricted by the fact that the covariates are assumed to lie on a 1/2^J spaced grid, even though they may not be equi-spaced. In addition to the univariate case, they extend their method to additive wavelets and further give rates of convergence for univariate case. The method performs well in simulations. Minor questions: - Can you show that the main optimization problem (3) has a unique solution? - Why not use the accelerated version of proximal gradient? - How does the interpolation approximation error 1/n \| f_0 - R \tilde{f}_0 \|_2^2 scale w.r.t K,n? - Please specify the interpolation matrix in your simulations. Typos: -Definition of \phi_{jk} in (1) is given in terms of \psi -Line 140: "Mallat", please give the relevant reference also -Line 143: O(K\log K), not O(K\log k). You mean, the top eigenvalue I guess, not all the eigenvalues? -Line 177: ca -> can (run a spell-checker?) -Line 279: existing